# Targeting mGlyR with nanobodies for depression

Thibaut Laboute [1], Stefano Zucca [2,10], Omar K. Sial[2,10], Mansi Sharma [3,10], Gloria Brunori[4,10], Shikha Singh [5,9], KV Nageswar[3], Haiyong Peng [6], Christoph Rader [6], Jérôme AJ Becker [1,7], Julie Le Merrer [1,7], Appu K. Singh [3,8] ✉ & Kirill A. Martemyanov[4] ✉

Development of therapies for neuropsychiatric conditions is one of the greatest challenges of modern medicine. Common limitations of traditional small molecule drugs include poor efficacy, off-target side effects and difficult druggability of many targets. In this study, we report a different approach deploying small engineered single domain antibodies, known as nanobodies, for the treatment of depression, a prevalent neuropsychiatric condition. We develop highly selective nanobodies for a recently discovered glycine receptor mGlyR crucially linked to pathophysiology of depression. Using a mouse model of stress-induced depression, we show that non-invasive intranasal delivery of nanobody produces rapid and lasting anti-depressant effect. We solve an atomic structure of mGlyR bound to nanobody and use a variety of cell-based approaches to reveal the mechanism of mGlyR modulation and its impact on neural circuitry. These findings support development of biologics for the treatment of intractable brain disorders.

Major depressive disorder (MDD) is a prevalent neuropsychiatric condition that affects nearly 5% of the population in developed countries[1]. While there has been tremendous progress in treating depression, the efficacy of currently approved therapeutics is limited[2]. Exploring strategies and drug targets has been deemed a relevant goal for managing MDD[3–5].

The molecular etiology of MDD is complex and not fully understood. Disbalance in neurotransmitter signaling is thought to trigger a range of maladaptive changes involving ion channels, kinases and second messengers, in particular cAMP, influencing synaptic communication and neuronal excitability[6,7]. These changes are precipitated by environmental factors such as stress, a major aggravating factor in MDD[8]. The neuronal circuits underlying the processing of emotional states and impacted by MDD are similarly complex and involve many structures, including the prefrontal cortex (PFC), a region heavily implicated in affective disorders and the effects of stress[9].

MDD treatment has traditionally focused on the elements that mediate signaling by monoamine neurotransmitters targeted by a vast majority of currently approved antidepressants[10]. However, several

[1]Université de Tours, INSERM, Imaging Brain & Neuropsychiatry iBraiN U1253, Tours, France. [2]Department of Neuroscience, The Herbert Wertheim UF Scripps Institute for Biomedical Innovation & Technology, University of Florida, Jupiter, FL, USA. [3]Department of Biological Sciences and Bioengineering, Indian Institute of Technology Kanpur, Kanpur, Uttar Pradesh, India. [4]Department of Physiology and Biophysics, University of Miami Miller School of Medicine, Miami, FL, USA. [5]Department of Biological Sciences, Columbia University, New York, NY, USA. [6]Department of Immunology and Microbiology, The Herbert Wertheim UF Scripps Institute for Biomedical Innovation & Technology, University of Florida, Jupiter, FL, USA. [7]Physiologie de la Reproduction et des Comportements, Unité Mixte de Recherche Centre National de la Recherche Scientifique 7247, Institut National de Recherche pour l'Agriculture, l'Alimentation et l'Environnement 0085, Institut National de la Santé et de la Recherche Médicale, Université de Tours, Nouzilly, France. [8]Mehta Family Centre for Engineering in Medicine, Indian Institute of Technology Kanpur, Kanpur, Uttar Pradesh, India. [9]Present address: BRIC-Translational Health Science and Technology Institute, Faridabad, Haryana, India. [10]These authors contributed equally: Stefano Zucca, Omar K. Sial, Mansi Sharma, Gloria Brunori. ✉e-mail: singhappu@iitk.ac.in; kmartemyanov@miami.edu

recent medications also target GABA, glutamate and opioid receptors, highlighting the potential of other neurotransmitter systems in developing MDD treatments[11,12]. One such untapped system involves the neurotransmitter glycine. It is released by specific neurons and has distinct effects on neural circuits and the activity of neurons[13,14]. Glycine and its related naturally occurring compound taurine have been heavily implicated in mood regulation and depression[15–17].

The effects of glycine had been thought to be mediated mainly by dedicated glycine receptor GlyR, an inhibitory ion channel[18]. Recently, a metabotropic receptor for glycine was discovered – mGlyR[19]. Formerly known as an orphan receptor GPR158, it exerts excitatory effects on neurons via modulation of the second messenger cAMP. The mGlyR is prominently expressed in the PFC, and its expression is regulated by stress[20,21]. The levels of mGlyR are markedly upregulated in patients diagnosed with MDD, and its knockout in mice produces anti-depressant phenotype and stress resilience[20]. The mGlyR employs an extracellular Cache domain for glycine recognition[19]. The binding of glycine or taurine to the ligand-binding pocket in this domain changes the activity of the associated Regulator of G Protein Signaling (RGS) complex at the intracellular side. This influences G protein signaling to second messengers and ion channels, thereby eliciting cellular response[19]. Notably, loss of RGS regulation also produces anti-depressant effects in mouse models[22]. Together, these observations point to mGlyR as an attractive target for developing anti-depressant therapies.

In this study, we explore the therapeutic relevance of targeting mGlyR as an antidepressant treatment. We develop a specific nano-body (Nb20) targeting the ligand-binding Cache domain of mGlyR, which mimics the effects of glycine. Through a series of studies, we show that Nb20 alters mGlyR function and its effects on neural circuits producing powerful antidepressant effects in mouse models when delivered in a non-invasive fashion.

## Results

### Development and characterization of nanobodies targeting mGlyR

Given the complete lack of selective chemical probes for mGlyR, we sought to obtain small protein ligands to alter mGlyR activity. The presence of extensive extracellular elements in mGlyR makes such a strategy attractive, considering recent success with other class C GPCRs[23,24]. We chose to generate single-domain antibodies (nano-bodies) for their high affinity towards targets and emerging potential for therapeutic translation. Phage library prepared from a llama immunized with recombinant mGlyR was screened using HEK293 cell membranes containing mGlyR (Fig. 1a). After three rounds of enrich-ment and rescreening, 61 individual clones were chosen. Corre-sponding nanobodies were isolated following expression in *E.coli* and tested for mGlyR binding using a flow cytometry strategy (Fig. 1b). Three clones showed positive interaction with mGlyR-expressing cells (Supplementary Fig. 1). Of these, clone number 20 (Nb20) showed the most robust signal and was chosen for further studies.

First, we have used flow cytometry to characterize Nb20 binding to mGlyR. Cells transfected with Venus-tagged full length mGlyR were incubated with purified myc-tagged Nb20 and its interaction with cells was monitored by APC-conjugated antibodies against myc (Fig. 1b). Using this approach, we were able to detect robust labeling of majority of mGlyR-transfected cells with Nb20 when excess Nb20 was used in the assay (Fig. 1c). Control experiments with cells lacking mGlyR, Nb20 or both showed no labeling of cells with Nb20 (Fig. 1c, Supplementary Fig. 2a). Titration experiments with increasing concentrations of Nb20 showed saturable profile with $EC_{50}$ of ~10 nM (Fig. 1d, e). The addition of mGlyR ligand glycine did not affect the binding of Nb20 with mGlyR (Supplementary Fig. 3). This interaction was specific to mGlyR, as we observed no Nb binding with other GPCRs, including related GPR179 and unrelated D1R (Supplementary Fig. 2b).

Next, we characterized the binding using surface plasmon reso-nance (SPR). In these experiments, we purified the recombinantly expressed extracellular portion of mGlyR (Ecto-mGlyR) tagged with an Fc tag and immobilized on the SPR chip via a mouse anti-human IgG CH2 monoclonal antibody (Fig. 1f). Applying increasing concentrations of Nb20 showed robust binding to the immobilized Ecto-mGlyR with an estimated $K_D = 375$ nM ($\chi^2 = 1.07$; $K_{ON} = 4.19 \times 10^4$ ms$^{-1}$ and a $K_{OFF} = 1.57 \times 10^{-2}$ s$^{-1}$ (Fig. 1g). The higher observed affinity of Nb20 in the cell-based applications, as compared to experiments with purified recombinant proteins, may be caused by an avidity effect that effec-tively enhances the binding to the low nanomolar level. Together, these data support the development of a selective affinity tool for mGlyR- Nb20.

### Nb20 inhibits mGlyR signaling via RGS7/Gβ5 complex

To study the functional consequences of mGlyR interaction with Nb20, we analyzed the ability of mGlyR to modulate the activity of the GTPase-activating Protein (GAP) complex RGS7/Gβ5, through which mGlyR transduces its signals. To monitor GAP activity of the mGlyR complex, we studied its impact on the kinetics of G protein deactiva-tion using a Bioluminescence Resonance Energy Transfer (BRET) assay (Fig.2a). This assay monitors changes in BRET signal upon interaction of Venus-Gβγ subunits with the masGRK3CT-Nluc reporter. Upon Gα deactivation, Venus-Gβγ dissociates from masGRK3CT-Nluc reporter, quenching the signal. As previously reported[19], we found that the introduction of RGS7/Gβ5 accelerated deactivation of its substrate, Gαo (Fig. 2b, d). Application of Nb20 had no significant effect on either baseline Gαo deactivation or RGS7/Gβ5-assisted process (Fig. 2b, d). However, when mGlyR was co-expressed together with RGS7/Gβ5, Nb20 significantly decelerated Gαo deactivation (Fig. 2c, d), suggest-ing that it specifically inhibited the GAP activity of RGS7/Gβ5 through mGlyR. Dose-response studies showed that the $IC_{50}$ of Nb20 on mGlyR is ~6 nM (Fig. 2e). In summary, these studies show that Nb20 serves as a selective inhibitor of GAP activity mediated by the mGlyR complex.

### Structural basis of mGlyR regulation by Nb20

To gain insights into the mechanisms underlying Nb20-mediated modulation of mGlyR, we obtained a high-resolution structure of mGlyR in complex with Nb20 using cryogenic electron microscopy (CryoEM) both with and without RGS7-Gβ5 complex (Fig. 3a, b). Three-dimensional classification (3D) of the particles revealed two dominant 3D classes, Nb-20-mGlyR and Nb-20-mGlyR-RGS7/Gβ5. Subsequently, these classes were refined to resolutions of 3.49 Å and 3.89 Å, respectively, without applying any symmetry (Supplementary Fig. 4, Supplementary Fig. 5). To improve the map, local refinements using a soft mask corresponding to ECD-Nb20 and TM-RGS7/Gβ5 were per-formed. The quality of the obtained maps exhibited higher resolution features in the TM region but relatively low resolution in the ECD and Nb20 binding regions. Nevertheless, these maps allowed us to con-struct the complete models of the mGlyR-Nb20 and mGlyR-RGS complexes, guided by known structures of mGlyR and the nanobodies (Supplementary Fig. 6).

We found that Nb20 was bound to the lateral side of the dimeric interface formed by two ligand binding Cache domains of the mGlyR dimer (Fig. 3c). The binding interface is predominantly mediated by the complementarity-determining regions 1 (CDR1) and possibly 2 (CDR2) of Nb20, establishing extensive contacts with the α2 helix and the loop between the β2 and β3 strands (residues 195–200) of subunit A, as well as a loop between the α1 and α2 helices (residues 140–153) of subunit B of mGlyR (Fig. 3d). The CDR2 of Nb20 interacts with a groove located at the dimeric interface between the two cache domains of the mGlyR receptor. Within the CDR2, the residue W58 lies in close spatial proximity to W162 from the α2 helix of subunit A of mGlyR, suggesting a potential interaction between these residues. Additionally, polar contacts may occur between R57 of Nb20 and E166 of subunit A, as well

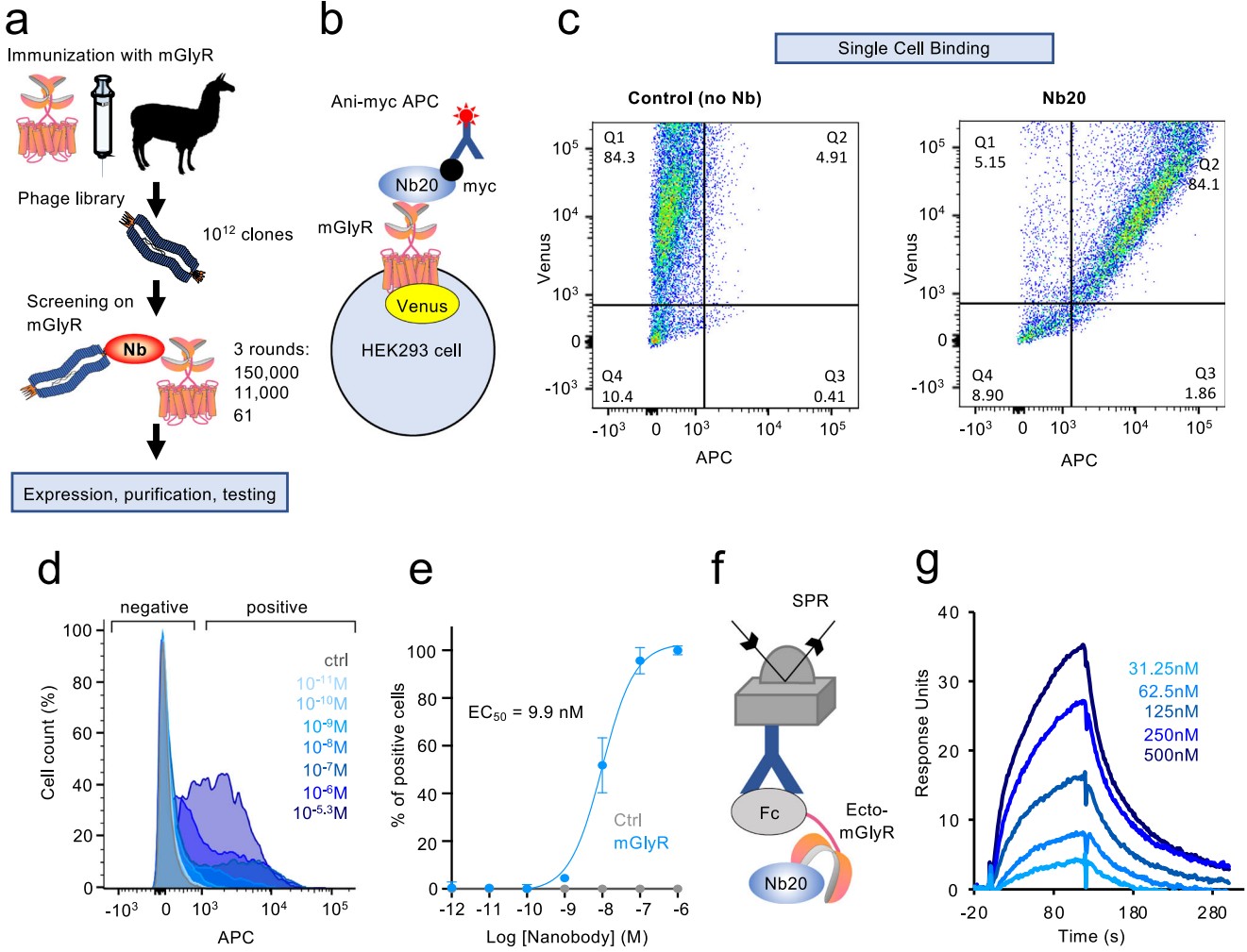

**Fig. 1 | Development of mGlyR-selective nanobodies. a** Schematic of the nanobody development pipeline. A phage library was constructed from leukocytes of a llama immunized with mGlyR, followed by 3 rounds of panning on mGlyR to enrich for specific binders, which were isolated and tested. **b** Schematic of the detection strategy in flow cytometry experiments. **c** Analysis of nanobody binding by flow cytometry of HEK cells transiently expressing mGlyR incubated with or without Nb20 and anti-myc-APC conjugated antibody. Percentages of cells in each quadrant are indicated. **d** Dose-response profiles of representative Nb20 binding experiment to cells expressing mGlyR in flow cytometry experiments. Concentrations of Nb20 are shown. **e** Quantification of data in panel D. Error bars are SEM values ($n = 3$ independent experiments). **f** Schematic of the surface plasmon resonance (SPR) assays that detect Nb20 binding to the chip containing immobilized recombinant ectodomain of mGlyR (Ecto-mGlyR). **g** SPR sensorgram of binding and dissociation of Nb20.

as between the backbone carbonyl of T59 Nb20 and N143 of the neighboring subunit of mGlyR (Fig. 3d). However, given the relatively low local resolution of the Nb20 and ECD regions, precise placement of side chains was not possible, and therefore these potential interactions should be interpreted with caution. We validated the binding mode of Nb20 by mutating and replacing all 6 critical residues in CDR1 and 9 residues in CDR2 regions with alanine and glycine (see Materials and "Methods"). This resulted in a complete loss of interaction of mutant Nb20 (Nb20*) with mGlyR (Supplementary Fig. 7). However, we can not exclude the possibility that these mutations may also affect the nanobody more globally. Notably, the ligand binding pocket is located adjacent to the binding groove of Nb20 (Supplementary Fig. 8a, b).

To determine whether binding to Nb20 indeed induces conformational changes in mGlyR, we first compared the Nb20-mGlyR structure to the apo structure of mGlyR alone. Global untethered superimposition of mGlyR-Nb20 on mGlyR-apo revealed conformational changes across various domains, including the Cache, stalk, and transmembrane (TM) domains (Supplementary Fig. 8c–e). Notably, the stalk domain exhibited pronounced movements, with deviations of up to 4.2 Å (Supplementary Fig. 8d). We next compared the transmembrane (TM) region in Nb20-bound complex structure with that of the

two previously determined structures of mGlyR–RGS7–Gβ5 without Nb20 (PDBs: 7SHF and 7EWP). While the overall TM architecture is conserved across structures, Nb20-structure showed notable deviations in TM3–TM4 and the connecting loop, ranging from relatively minor -1.1 Å (7EWP) to substantial -5.5 Å (7SHF) (Supplementary Fig. 8f, h).

To pinpoint the precise conformational changes in the extracellular domain induced by Nb20 binding, we aligned the transmembrane (TM) domains (Fig. 3e). Anchoring the TM that way allowed us to observe dramatic conformational changes in the extracellular domain (ECD) upon Nb20 binding to the receptor. Specifically, in the Nb20-bound structure, the ECD translated by as much as 12 Å and rotated by approximately 7° in relation to the mGlyR-apo structure (Fig. 3f). These observations suggest that binding to Nb20 starts the wave of events that eventually remodel the cytoplasmic interface of the receptor involved in interaction with RGS7/Gβ5 and transduction of the signal. To test this model, we further compared the cryoEM structure of Nb20-mGlyR-RGS7/Gβ5 with previously solved mGlyR-RGS7/Gβ5 structures (PDB: 7SHF, 7EWP) (Fig. 3g, Supplementary Fig. 8f, h). In the mGlyR-RGS7/Gβ5 structures, the ECD domain was highly flexible, resulting in low-resolution features as observed in 7EWP[25] or scattered

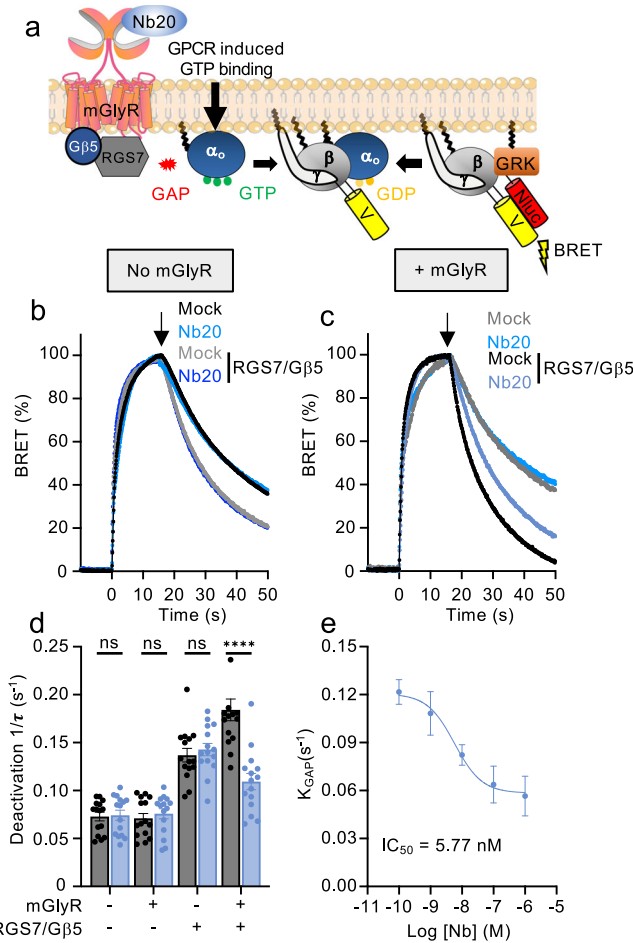

**Fig. 2 | Regulation of GAP activity of mGlyR - RGS7/Gβ5 complex by Nb20.**
**a** Schematics of the BRET-based GAP assay to study the activity of mGlyR. First, Gαo is activated by the dopamine D2R receptor. After reaching steady state, D2R antagonist haloperidol is injected, and the kinetics of G protein deactivation are monitored by following the quenching of the BRET signal. **b**, **c** Traces of BRET signal showing Gαo activation and deactivation time course with or without Nb20 treatment in cells without mGlyR (**b**) or cells transfected with mGlyR (**c**).
**d** Quantification of deactivation time constant of the reactions presented in panels B and C. $1/\tau$ is calculated from deactivation curves of $n = 5$ independent experiments conducted in triplicate from each cell transfection group. Data represent mean ± SEM. ****$p < 0.0001$, ns (not significant) = $p > 0.05$ ($p = 0.99$, $p = 0.98$ and $p = 0.96$ for no mGlyR no RGS7-Gβ5, mGlyR only and RGS7-Gβ5 only, respectively, two-way ANOVA. **e** Dose–response profile of changes in GAP activity ($K_{GAP}$) calculated by subtracting the baseline deactivation rate ($1/\tau$) from the rate of the reaction in the presence of mGlyR-RGS7-Gβ5. Data are mean ± SEM of $n = 3$ independent experiments conducted in triplicate.

density at relatively higher resolution in 7SHF[26] structures. The binding of Nb20 appeared to improve ECD stability in the presence of the bound RGS7-Gβ5 module, yielding a relatively well-resolved density for ECD. Strikingly, comparison revealed that the density corresponding to the RGS domain, which typically wraps around Gβ5 from below, was poorly resolved in the Nb20-bound structure, suggesting that the RGS domain adopts a more flexible conformation compared to mGlyR-RGS7/Gβ5 structures, where this region was resolved at higher resolution. (Fig. 3b, Supplementary Fig. 8f, h). Even after local refinement and analysis at various contour levels, the RGS domain density remained poorly resolved, suggesting that this domain becomes highly flexible upon binding the Nb20 to ECD. Superimposition of Nb20-bound and free structures by the 7TM region suggested plausible changes. The DEP/DEX domain of RGS7, which directly contacts

the receptor, adopted a distinct conformation in the Nb20-bound structure. Notably, the β3-hairpin loop becomes unresolved (Fig. 3g), which may affect the RGS complex. Nb20 binding appeared to shift CTH2 by 2.2 Å, induce a 1.2 Å displacement in RGS7, and 1.5 Å movement in the TM3-TM4 connecting loop, and up to a 5.5 Å (when compared with 7SHF) changes in the TM3, which serve as the contact site for RGS7 (Supplementary Fig. 8f, l). However, the exact side-chain orientation at these regions could not be resolved due to the low local resolution, limiting the structural analysis to the level of secondary structure elements. A high-resolution structure of RGS7-Gβ5 will be required to delineate the remodeling of hydrogen bonding and hydrophobic interactions following Nb20 binding. Although the Gβ5 in the Nb20-bound structure appears to shift by approximately 5 Å compared to its position in the structure without Nb20, this displacement should be interpreted cautiously, given the poorly resolved density for Gβ5 (Fig. 3g, Supplementary Fig. 8g). Together, structural studies suggest a model where binding of Nb20 to the extracellular ligand binding Cache domain of mGlyR may promote specific stabilized conformation of the receptor which possibly remodels its intracellular interaction with the RGS7/Gβ5, leading to a higher flexibility of its catalytic RGS domain, decoupling it from regulating Gα proteins, e.g. by changing its orientation towards the membrane where Gα protein are residing.

## Nb20 produces anti-depressant effects in mice

Our functional data indicate that Nb20 blocks the ability of mGlyR-RGS7/Gβ5 complex to regulate G protein signaling. Previous studies indicated that genetic deletion of either mGlyR[20] or RGS7[22] produced a substantial antidepressant phenotype in mice and stress resilience. Therefore, we have next tested whether administration of Nb20 in vivo would have similar behavioral effects. Sequence analysis of llama-derived Nb20 reveals favorable immunogenicity profile likely making it suitable for in vivo applications in other species (Supplementary Fig. 9). Mice were injected, delivering either Nb20 or vehicle control into the brain (Fig. 4a) and evaluated in a panel of behavioral tests that generally assess various aspects of affective-like states 24 h later (Fig. 4b). We found that mice that received Nb20 buried fewer marbles in the marble burying test, had reduced immobility in the tail suspension and forced swim tests relative to control animals but their behavior in the elevated plus maze did not reach threshold for statistical significance when compared to untreated mice (Fig. 4b). Calculating overall emotionality score based on multiple measures confirmed that mice treated with Nb20 displayed prominent antidepressant-like phenotype (Fig. 4c). Remarkably, these behavioral differences between the groups persisted for at least two weeks following the treatment at which point anxiolytic effect slightly increased (Supplementary Fig. 10). To confirm the specificity of the effects, we used mutated Nb20* incapable of binding to mGlyR and found that treatment of mice with these nanobodies did not produce significant effects in any of the paradigms used (Supplementary Fig. 11). We further conducted a panel of tests to measure the effect of Nb20 extending the range of behavior al testing to better understand Nb20 actions. We found that treatment of mice with Nb20, but not inactive Nb20*, alleviated novelty-induced hypophagia, an approach behavior influenced by depression/anxiety-related affective states (Supplementary Fig. 12a). Interestingly, Nb20 further enhanced an ability of mice to recognize novel object suggestive of the positive effect in the cognitive domain (Supplementary Fig. 12b). Importantly, Nb20 treatment had no effect on locomotor activity (Supplementary Fig. 13a) or motor performance (Supplementary Fig. 13b) of mice.

To further explore the therapeutic utility of Nb20, we assessed its effects in a stress-induced depression model where mice are exposed to a chronic variable stress (Fig. 4d, Supplementary Fig. 14). In this study, Nb20 was also administered non-invasively through intranasal delivery, given the translational relevance of this method for treating

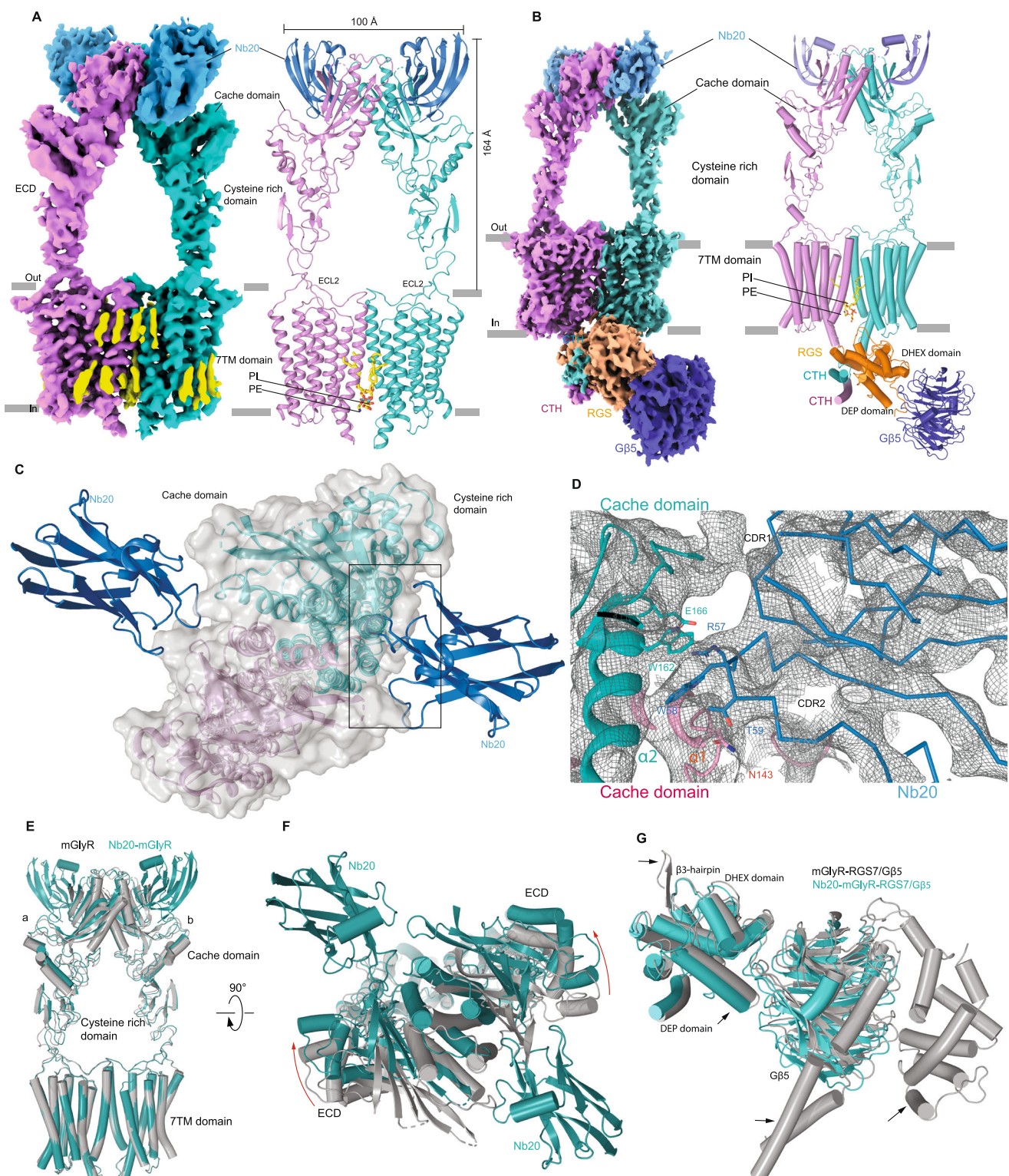

**Fig. 3 | CryoEM structure of the mGlyR in complex with Nb20. a** Side view of the cryo-EM map (left) and corresponding model (right) of the mGlyR-Nb20 homo-dimer, with individual monomers depicted in teal and violet. Phospholipids and cholesterols are represented in yellow. **b** Cryo-EM map (left) and corresponding model (right) of the Nb20-mGlyR-RGS7-Gβ5 homodimer, with monomers colored as in (**a**). RGS7 is shown in orange, while Gβ5 is depicted in blue. **c** Top view of Nb20-bound Cache domain of mGlyR. **d** Detailed interaction network between Nb20 and the Cache domain with EM density shown as mesh in gray. **e** Side view of the TM-based structural superimposition of mGlyR-apo structure depicted in gray with that of mGlyR-Nb20 structure shown in teal. **f** A 90° rotation relative to the left panel (**e**) showing a top view of the structural comparison of the Cache domain. **g** Structural superimposition of RGS bound to mGlyR-apo and Nb20-mGlyR.

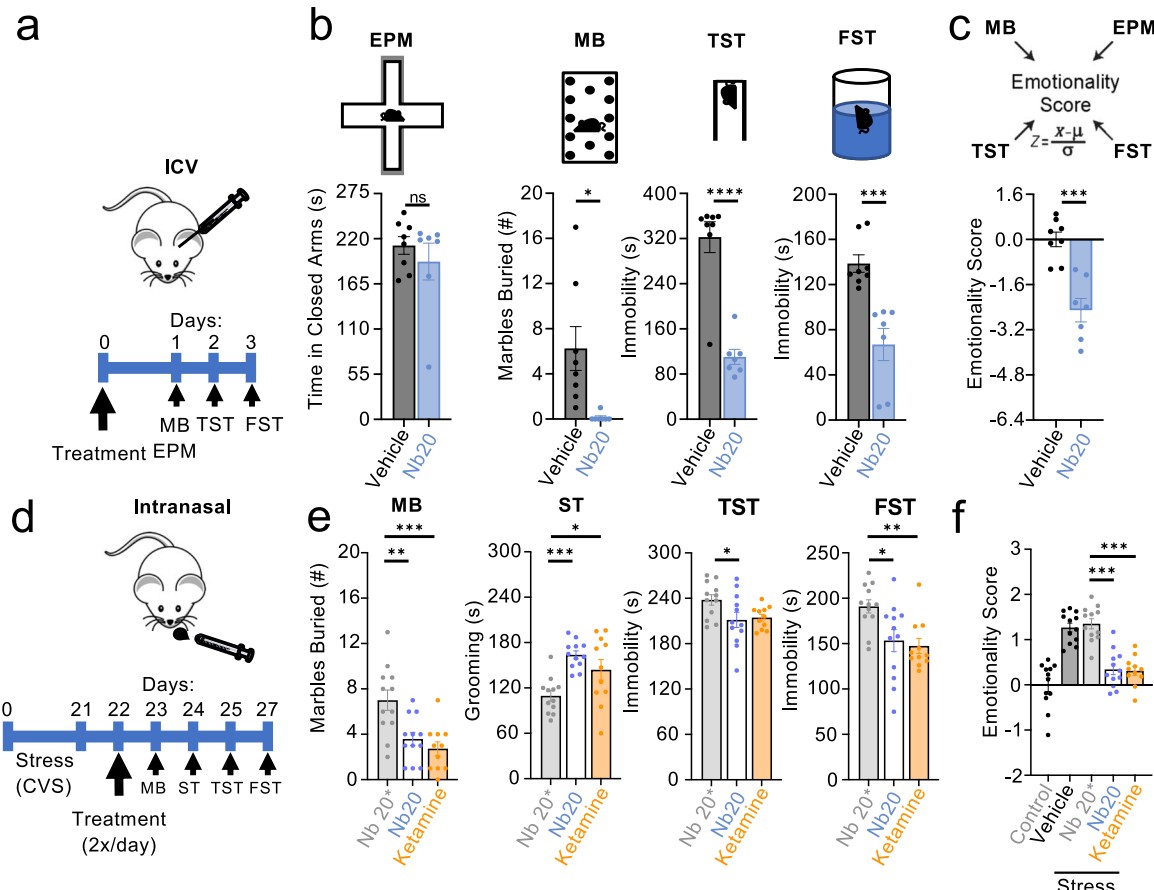

**Fig. 4 | Antidepressant effects of Nb20 administration in mice. a** Schematic of the intracerebroventricular (ICV) injection strategy for the administration of vehicle or Nb20 to mice. **b** Evaluation of mice injected with Nb20 or vehicle control in a panel of behavioral tests consisting of elevated plus maze (EPM), marble burying (MB), tail suspension test (TST), and forced swim test (TST) ($n = 8$ mice in vehicle (4 males and 4 females) group and 7 (4 males and 3 females) in Nb20 treated). Data are mean ± SEM (Unpaired $t$-test; ns $p = 0.4259 > 0.05$, *$p = 0.0119 < 0.05$, ***$p = 0.0005 < 0.001$, ****$p < 0.0001$). **c** Calculation of emotionality scores based on superscoring of four behavioral tests. ($n = 8$ mice in vehicle and 7 mice Nb20 treated) Data are mean ± SEM (Unpaired $t$-test; ***$p < 0.001$). **d** Schematic of the intranasal delivery for the administration of ketamine, Nb20 or control Nb20* to mice, timeline and experimental strategy for stress induction, treatment and

behavioral evaluation. **e** Evaluation of mice in behavioral paradigms as indicated ($n = 12$ mice/group: 6 males and 5–6 females). Data are mean ± SEM (One-way ANOVA; Tukey's test, MB: $F_{(2,32)} = 10.02$, $p = 0.004$, Nb20*-Nb20 **$p = 0.0047 < 0.01$, Nb20*-ketamine ***$p = 0.0006 < 0.001$, ST: $F_{(2,32)} = 9.605$, $p = 0.0005$, Nb20*-Nb20 *$p = 0.0293 < 0.05$, Nb20*-ketamine ***$p = 0.0004 < 0.001$, TST: $F_{(2,32)} = 3.655$, $p = 0.0372$, Nb20*-Nb20 *$p = 0.0476 < 0.05$, FST: $F_{(2,32)} = 6.146$, $p = 0.0055$, Nb20*-Nb20 *$p = 0.0215 < 0.05$, Nb20*-ketamine **$p = 0.0086 < 0.01$). **f** Calculation of emotionality scores based on superscoring of four behavioral tests ($n = 12$ mice/group: 6 males and 6 females) for all treatments in comparison with non-stress control and vehicle-treated stressed mice. Data are mean ± SEM (One-way ANOVA; Tukey's test, <0.0001, Nb20*-Nb20 ***$p < 0.0001$, Nb20*-ketamine ***$p < 0.0001$).

depression[27–29]. Furthermore, the effects were benchmarked to the effects of rapid anti-depressant treatment with ketamine delivered the same way (Fig. 4d, e). Remarkably, we found that intranasal delivery of Nb20 produced a rapid and powerful antidepressant effect in stressed mice (Fig. 4F) with similar effects in males and females (Supplementary Fig. 15). The extent of this effect matched the antidepressant effects of ketamine across all behavioral paradigms. We did not detect the effect of Nb20 on either anxiety-related behaviors of stress-exposed mice or general motor activity (Supplementary Fig. 16). Neither did we observe Nb20 influencing the hedonic component in the sucrose preference task (Supplementary Fig. 16). Furthermore, Nb20 was able to normalize stress-induced elevation of corticosterone, prominent in female mice (Supplementary Fig. 17). Together, these results indicate that Nb20 produces antidepressant-like effects in a mouse model of stress-induced depression.

### Inhibition of mGlyR with Nb20 modulates physiological properties of mPFC neurons

We finally sought to assess the impact of Nb20 on the activity of ex vivo intact neural networks by examining the intrinsic properties of layer II-

III neurons in the prelimbic cortex, where mGlyR is prominently expressed[29]. Knockout of mGlyR or its inhibition with endogenous ligand glycine has been shown to increase the excitability of pyramidal neurons in layer II-III[28], among other effects on synaptic physiology, which vary across different brain regions and neuronal populations[30–32]. Thus, blockade of mGlyR with Nb20 would be predicted to increase the excitability of layer II-III neurons. To test this hypothesis, we incubated brain slices with Nb20 for 10 min prior to recordings and compared excitability of layer II-III neurons with the untreated slices or slices incubated with control Nb20* nanobody deficient in mGlyR binding (Fig. 5a). Pre-incubation with Nb20 significantly increased the number of action potentials fired over the ramp while decreasing the amount of current necessary to elicit the first action potential (Fig. 5b–d) without changes in the resting membrane potential (Fig. 5e, Supplementary Fig. 18). In contrast, control Nb20* had no effect on neuronal firing, rheobase current or resting membrane potential (Fig. 5b–d, Supplementary Fig. 18). Together, these results indicate that inhibition of mGlyR with Nb20 increases excitability of layer II-III neurons in prelimbic cortex, in a manner similar to genetic loss of mGlyR or its inhibition by glycine.

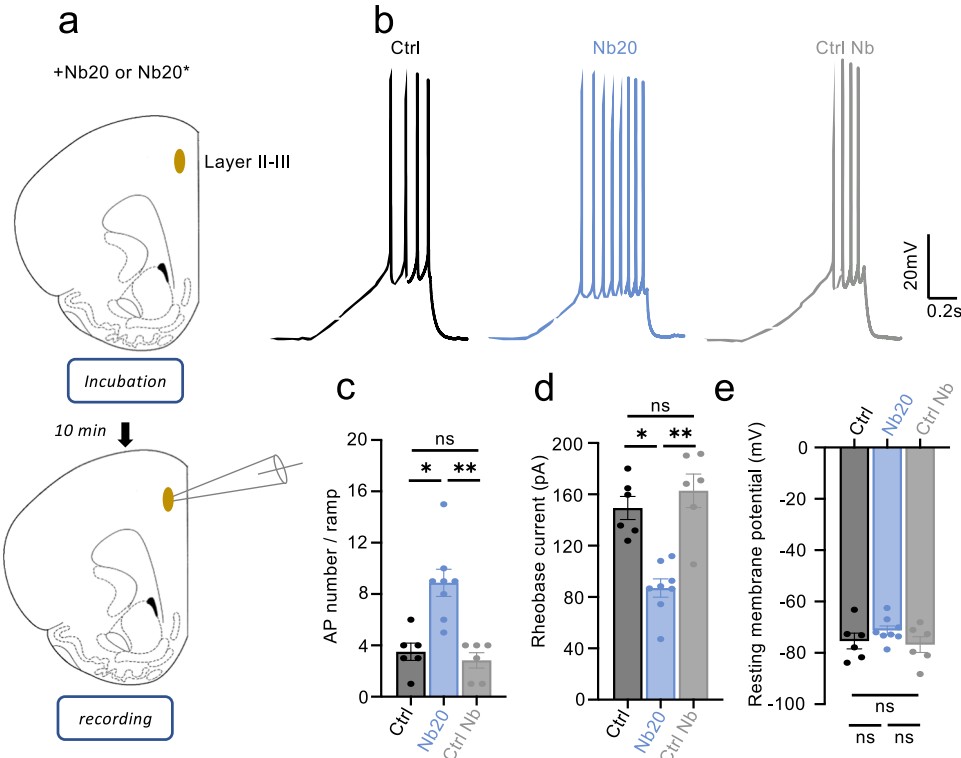

**Fig. 5 | Effect of Nb20 on neuronal excitability. a** Schematic of the electro-physiological recordings in slice preparation targeting mPFC neurons of layer II-III in WT mice. Slices were incubated with ACSF (Ctrl), Nb20, or mutated Nb20* before recording **b** Representative traces of voltage responses to a 200 pA current ramp injection under different conditions. **c** Quantification of changes in excitability by number of action potentials fired in response to 200 pA current ramp ($n = 6$ neurons in control group, $n = 8$ neurons in Nb20-treated group, $n = 6$ neurons in Nb20* treated group). Data are mean ± SEM. Nonparametric One-way ANOVA; Kruskal-Wallis test, **$p = 0.0032 < 0.01$, *$p = 0.0122 < 0.05$ and ns = $p > 0.9999 > 0.05$.

**d** Quantification of changes in excitability by rheobase current ($n = 6$ neurons in control group, $n = 8$ neurons in Nb20-treated group, $n = 6$ neurons in Nb20* treated group). Data are mean ± SEM. Nonparametric One-way ANOVA; Kruskal-Wallis test, **$p = 0.0028 < 0.01$, *$p = 0.0217 < 0.05$ and ns = $p > 0.9999 > 0.05$. **e** Resting membrane potential of layer II-III pyramidal neurons in WT mice ($n = 6$ neurons in control group, $n = 8$ neurons in Nb20-treated group, $n = 6$ neurons in Nb20* treated group). Data are mean ± SEM. One-way ANOVA; Kruskal-Wallis test, ns = $p > 0.05$ (respectively: Ctrl vs. Nb20 $p = 0.6608$, Ctrl vs. Nb20* $p > 0.9999$, Nb20 vs. Nb20* $p = 0.7862$).

## Discussion

In this study, we suggest that an immunotherapy solution could be potentially applied for the treatment of a major neuropsychiatric condition- major depressive disorder. Immunotherapies offer a high degree of specificity, low off-target and toxicity and high efficacy and have become a method of choice for treating cancer and autoimmune disorders[33,34]. Immunotherapies are also becoming increasingly applied for brain conditions, most successfully in managing neuro-degenerative conditions[35]. Intriguingly, a few recent studies also indicate their potential utility for managing neuropsychiatric conditions[36–39]. In this work, we applied an immunotherapy strategy to effectively suppress depression-related behaviors in mice, thereby uncovering a fresh direction for therapeutic interventions for a significant health crisis. We recognize the limitations of modeling neuropsychiatric disorders in rodents. Ultimately, translation of our findings would require sequence optimization of the Nb20 to increase its brain penetrance, reduce antigenicity, as well pharmacokinetic studies finding preferred administration route before tolerability and efficacy clinical trials in humans. We further recognize that more work is required before our observations could be translated to humans, especially in the domain of characterizing the toxicity of the Nb20 as well as their on-target and potential off-target effects on neuronal physiology, as well as pharmacokinetic properties.

The key to our approach was to develop a single domain antibody, known as a nanobody, recognizing a discovered metabotropic receptor for glycine, mGlyR, as a target for developing antidepressants[19]. Its genetic knockout in mice results in a marked antidepressant phenotype and resilience to stress-induced depression[20]. Molecularly, mGlyR acts non-canonically by engaging the RGS7/Gβ5 complex rather than typical G proteins to transduce its signals. Glycine acts as a suppressor of mGlyR signaling, thereby relieving the inhibitory influence that RGS7/Gβ5 imposes on G proteins, disinhibiting them and allowing signal propagation[19]. Consistent with the inhibitory influence of glycine on mGlyR and mGlyR knockout being antidepressant, we find that our nanobody, Nb20, also inhibits mGlyR and produces an anti-depressant effect. While we observe that Nb20 increases the excitability of layer II-III neurons in the PFC via mGlyR, the causality of this effect is uncertain. Nb20 likely modulates mGlyR on multiple neurons and regulates several processes to exert the anti-depressant effects and reverse the effects of stress. The contribution of mGlyR to these processes are not well defined and remains controversial[20,30–32]. Similarly, stress exerts a plethora of complex effects on neurophysiology by altering the expression and function of a number of molecules across the brain[40,41]. How inhibition of mGlyR by Nb20 alters the effects of stress on the brain circuitry to produce anti-depressant effects will require understanding the mechanisms by which stress impacts affective states, which remain incompletely understood. Defining the effects of Nb20 on the physiological properties of neurons and their responses to stress will be an important area for future exploration. Further work will also be needed to rule out the contribution of other targets which might be unintentionally engaged by Nb20 to produce anti-depressant effects.

Detailed structural and mechanistic studies show that Nb20 interacts with the ligand-binding Cache domain of mGlyR and inhibits

the GAP activity of RGS7/Gβ5 towards its substrate Gαo. Thus, Nb20 serves as a synthetic antagonist specific for mGlyR. There is currently no structure of glycine-bound mGlyR, and future experiments will need to explore whether Nb20 taps into the mechanism of mGlyR antagonism utilized by its endogenous ligand glycine. Of further interest is the presence of the additional elements in the C-terminus of mGlyR, which play a critical role in allosteric potentiation of GAP activity of RGS7/Gβ5[42] but are not captured in our structure of the truncated receptor. Because Nb20 effectively inhibits full-length mGlyR and also induces prominent structural changes in the truncated mGlyR, it suggests that Nb20 inhibition, which results in repositioning of the RGS7/Gβ5, occurs independently of the allosteric potentiation conferred by the C-terminus of the receptor. Loss of the density for the catalytic RGS domain suggests increased flexibility of the region, which would disfavor regulation of its substrate Gαo strictly oriented on the membrane. How exactly Nb20 influences the conformation of the 7TM region and particularly its interface with RGS7/Gβ5 has not been fully resolved in this study, given the differences between the two previously reported ground state structures of mGlyR without Nb20. Additional structural work that captures intermediate conformations of the receptor complexes will be required to fully resolve the mechanism of RGS/Gβ5 modulation by mGlyR ligands such as Nb20 or glycine. It will also be interesting to determine the structural basis of allosteric potentiation of RGS7/Gβ5 by the C-terminus of mGlyR and how it is influenced by Nb20 and glycine. Future studies will also need to establish whether and how Nb20 and glycine affect alternative modes of RGS7/Gb5 engagement by mGlyR observed in other studies[25].

Over recent years, nanobodies have been widely deployed as reagents for studying and manipulating GPCRs in the nervous system[43–46]. They show several key advantages that make them superb tools, including unsurpassed target selectivity, the ability to recognize distinct conformational states, stability, and relatively small size, which facilitates target access[47]. As a result, nanobodies are increasingly adapted for therapeutic applications to brain disorders[39,48]. The exciting potential of their therapeutic utility is fueled by multiple reports that nanobodies can efficiently reach their targets in the central nervous system. Several mechanisms likely enable this, including their active and passive transport across the blood-brain barrier, transcytosis and carrier-assisted delivery[49–52]. Intriguingly, we show that Nb20 targeting mGlyR shows in vivo antidepressant efficacy when administered intranasally to mice. These observations are similar to previous reports of effective intranasal nanobody delivery[53,54] and suggest that they are readily uptaken by the olfactory neurons. Furthermore, the delivery of nanobodies may be further augmented in depression-related states known to disrupt the blood-brain barrier[55]. Overall, our studies support the efficacy of non-invasive nanobody-based therapies for brain disorders, at least in rodents, and suggest that it could be explored as a modality for the treatment of depression, in particular if it is accompanied by cognitive issues and unencumbered by anxiety.

## METHODS
### Animals
All animal experiments were approved by the Herbert Wertheim UF Scripps Institute for Biomedical Innovation & Technology Institutional Animal Care and Use Committee (IACUC) as well as by the Comité d'Ethique en Expérimentation animale Val de Loire (C2EA-19; APAFIS #46226-2022022509329925 v6). All experimental procedures were conducted in accordance with guidelines by NIH and the European Communities Council Directive 2010/63/EU. Animal studies are reported in compliance with the ARRIVE guidelines. The *Gpr158*−/− mice were purchased from KOMP (*Gpr158*tm1(KOMP)Vlcg) and maintained on a C57/Bl6 background and bred as heterozygous pairs to generate *Gpr158*−/− and *Gpr158*+/+ littermates. After weaning, male and females

were separately group-housed under standard conditions in a pathogen-free facility on a 12:12 light:dark hour cycle with access to food and water *ad libitum*. For the chronic variable stress paradigm, wild-type male and female adult (8-week-old) C57Bl/6 J mice (Charles River, Massachusetts) were habituated for one week before experimental manipulation. These mice were single-housed at room temperature (-24 °C) under a 12-h light/dark cycle (07:00-19:00) with *ad libitum* access to water and food, except during testing.

### cDNA constructs
Dopamine D2 receptor (cDNA Resource Center: #DRD0200001), RGS7 (GenBank: AY587875), Gβ5 (GenBank: NM_016194), Gα$_{OA}$ (cDNA Resource Center: #GNA0OA0000) in pcDNA3.1(+) were purchased from cDNA Resource Center (https://www.cdna.org). masGRK3ct-Nluc, Venus-156-239-Gβ1, Venus-1-155-Gγ2 were synthesized by GenScript in pcDNA3.1 + . pCMV5 plasmid encoding GαoA was a gift from H. Itoh (Nara Institute of Science and Technology, Japan). GPR158 ectodomain (aa 1-417) was subcloned into a previously described Fc and 6xHis tagged vector[32]. Nanobody production was done using the pCANTAB vector (phage display) and promising candidates, including Nb20, were subcloned into pET28a. The complete sequence of Nb20 is as follows: MAEVQLQESGGGLVQAGGSLRLSCAASGSIGNIYIMG-WYRQTPGPQRELVATIRTVRWTKYE-DYADSVKGRFTISDDDAKNTVYLQMNSLKPEDTAVYYCNYKDY-NAPSDGYWGQGTQVTVSSEPKTPKPQ. To generate control non-binding nanobody (Nb20*), Nb20 cDNA was mutated to replace 30IGNIYI35 sequence in CDR1 with 30GGAGAG35 sequence and 54RTVRWTKYE62 sequence in CDR2 with 54GAVGGAAAG62 sequence using direct mutagenesis by PCR with 2 sets of primers: CDR1 Forward primer: GCGGCGGTGCTGGCGCTGGCATGGGCTGGTACCGCCAG; CDR1 Reverse primer: CGCCAGCACCGCCGCTTCCAGAGGCTGCACAGGAGAGTC and CDR2 Forward primer: CAGCCGCAGCCGCAGCCGCAGCCGACTATGCAGACTCCGTAAA GGGC; CDR2 Reverse primer: CTGCGGCTGCGGCTGCGGCTGCAGT TGCGACCAGCTCGCG.

### Chemical and drugs
The following chemicals were used. Dopamine hydrochloride (MilliporeSigma Cat# H8502), Haloperidol (MilliporeSigma Cat# H1512), Dulbecco's Phosphate-Buffered Saline (PBS) (Gibco Cat#10010-023) with 0.5 mM MgCl$_2$ and 0.1 % glucose, glycine (National Diagnostics # EC-405), Dulbecco's modified Eagle's medium (Thermo Fisher Scientific Cat# 11965-092), Fetal bovine serum (Genesee Scientific Cat# 25-550), Sodium pyruvate (Thermo Fisher Scientific Cat# 11360-070), MEM non-essential amino acids (Thermo Fisher Scientific Cat# 11140-050), Penicillin-streptomycin (Thermo Fisher Scientific Cat#15140-122), Matrigel (Corning Cat# 356230), METAFECTENE® PRO (RKP203/RK092820, Biontex Germany), Dulbecco's phosphate-buffered saline (MilliporeSigma Cat# D5652), PEG6000/2 M NaCl (Teknova #P4168), Nano-Glo luciferase (N113B Promega), Isopropyl β-D-1-thiogalactopyranoside (IPTG) (Sigma-Aldrich #I6758), Kanamycin (Thermo Fisher Scientific #11815032). The following chemicals were prepared: LEW (50 mM NaH2PO4, 300 mM NaCl, pH adjusted to 8.0 using NaOH), 2xYT (16.0 g/l tryptone, 10 g/l yeast extract, 5 g/l sodium chloride, Final pH 6.8 ± 0.2 at 25 °C, autoclaved).

### Llama immunization, phage display
One llama was immunized in strict accordance with good animal practices following the EU animal welfare legislation law with HEK293 cell membrane expressing mGlyR at Eurogentec (Belgium). Blood samples from immunized llama were harvested 87 days after immunization, and RNA from leukocytes was extracted using LeukoLOCK Total RNA Isolation system (Life Technologies #AM1923) accordingly to manufacturer instructions. Reverse transcription was performed on

extracted RNA, and cDNA was amplified by PCR using 8 pairs of primers designed to amplify the variable heavy only domains (VHH) of non-conventional IgG2 and IgG3 and introduce NotI and SfiI cleavage sites, respectively (in bold in the following sequence).

Forward primers:

VH_11: GTCGTC**GGCCCAGCCGGCC**ATGGCCGAGGTGCAGCTGGTGGAGTCTGGGGGAGG

VH_12: GTCGTC**GGCCCAGCCGGCC**ATGGCCGAGGTGCAGCTG-CAGGMGTCTGGGGGAGG

VH_14: GTCGTC**GGCCCAGCCGGCC**ATGGCCGAGGTGCAGCTG-CAGGCGTCTGG

VH_13: GTCGTC**GGCCCAGCCGGCC**ATGGCCGAGGTGCAGCTG-CAGGAGTCWGG

Reverse primers

VH_sh: GCTGCT**GCGGCCGC**GGGGTCTTCGCTGTGGTGCGC

VH_lg: GCTGCT**GCGGCCGC**TTGTGGTTTTGGTGTCTTGGG

Amplified cDNA was purified and digested with NotI and SfiI, following by ligation into the phagemid vector pCANTAB5 and electroporation of competent *E. coli* TG1 cells. Bacteria were grown in 2xYT broth containing 2% glucose at 37 °C under agitation until $OD_{600nm}$ reached 0.5. Subsequently, helper phage KM13 ($2 \times 10^{11}$ units) was added for 30 min at 37 °C without agitation. Infected bacteria were pelleted by centrifugation at 4000 g for 20 min at 4 °C, and the pellet was resuspended in 2xYT containing 120 µg/ml ampicillin and 50 µg/ml kanamycin. The phage library was grown overnight at 37 °C with agitation. Phages expressing a nanobody library were harvested and purified by centrifugation at 8000 g for 15 min at 4 °C, supernatant was precipitated using 20% of PEG6000/2.5 M NaCl for 1 h at 4 °C, followed by centrifugation at 8000 g for 15 min at 4 °C and pellet was resuspended in PBS-glycerol 15% and stored at +4 °C.

### Nanobody identification via phage library screening

To obtain anti-mGlyR-specific nanobody clones, two steps of depletion were first performed. Library containing $10^{12}$ phage units was incubated in the first well of a MaxiSorp plate (ThermoFisher Scientific #441653) coated with 100 µg of HEK293 cell membranes mock-transfected for 1 h at RT, and supernatant was transferred to a second identically coated well for 1 h at RT in order to reduce non-specific binders. A last step of selection was performed using the remaining phages incubated on one well coated with 100 µg of cell membranes expressing mGlyR for 2 h at RT. After washing, bound phages were eluted with Tris 50 mM pH = 8, 1 mM $CaCl_2$, Trypsine 1 µg/ml for 15 min at RT. Eluted phages were recovered and amplified with *E. coli* TG1 cells infected in 2xYT broth containing ampicillin and 2% glucose overnight at 30 °C with agitation. Next day, $2 \times 10^9$ units of helper phage KM13 (AKA VCSM13 Agilent Technologies #2002521) were added to the amplified TG1 cells for 1 h at 37 °C, followed by centrifugation at 3000 g for 10 min at RT. Pellet was resuspended in 2xYT broth containing 100 µg/ml ampicillin and 25 µg/ml kanamycin and incubated overnight at 30 °C with agitation. To separate TG1 cells and amplified phages, the overnight culture was centrifuged at 3000 g for 30 min at 4 °C and phages in the supernatant were precipitated for 30 min on ice using 20% PEG-6000/NaCl 2.5 M, followed by a centrifugation at 10,000 g for 10 min at 4 °C. Pellet was resuspended in PBS with 15% glycerol and stored at −20 °C, and the rest was used in the second round of panning selection as described previously. A total of 3 rounds of panning were performed to identify 61 individual clones. Each clone was produced directly from individual TG1 colonies induced by isopropyl-β−26-D-thiogalactopyranoside (IPTG) for flow cytometry screening.

### Protein production and purification

Individual nanobody cDNAs were fused with a c-myc and 8xhis tags at the C-terminus, subcloned into the pET28a vector using the in-fusion HD cloning kit (Takara Bio #102518) and sequenced. Freshly transformed *E. coli* BL21 DE3 (NEB # C2527H) with pET28a plasmid encoding nanobody was grown at 37 °C, shaking at 220 rpm in Terrific Broth (Kd Medical Inc #501018968) containing 50 µg/ml kanamycin until $OD_{280nm}$ = [0.6–0.8]. Protein expression was then induced by 1 mM Isopropyl β-D-1-thiogalactopyranoside (IPTG), and bacteria were grown overnight at 28 °C with shaking at 220 rpm. Bacteria were collected, centrifuged, and the pellet was lysed by sonication in LEW 1X at 4 °C. Periplasmic solution were harvested after centrifugation at 17,000 g, 30 min, 4 °C and purified on a Ni-IDA column (Macherey Nagel # 745160) according to manufacturer recommendations. Ecto-GPR158-Fc was produced in HEK293FT cells (supernatant) purified (Nickel column) dialyzed/concentrated (Amicon tube 30 kDa). Protein concentration in supernatant was determined using UV spectroscopy (Nanodrop).

### Cryo-EM sample preparation and data acquisition

The cryo-EM sample was prepared as described previously[26]. The protein sample, the purified Nb20-mGlyR-RGS7-Gβ5 complex, was prepared for cryo-electron microscopy (Cryo-EM) imaging. A total of 3.0 µL of the protein sample was applied to glow-discharged 200 mesh gold grids (UltraAufoil R1.2/1.3) inside an FEI Vitrobot Mark IV (Thermo Fischer Scientific). The Vitrobot was maintained at 4 °C with 100% humidity. Prior to blotting, a blot force of 2, a blot time of 2 s, and a wait time of 20 s were applied to remove excess sample. The grids were then plunge-frozen in liquid ethane to preserve their vitrified state.

Cryo-EM imaging of the Nb20-mGlyR-RGS7-Gβ5 protein complex was performed using a 300 kV Titan Krios electron microscope equipped with a Gatan K3 Summit direct electron detection (DED) camera (Gatan, Pleasanton, CA, USA) and a post-column GIF Quantum energy filter operating in counting mode. The microscope was calibrated to a magnification of 105,000, resulting in a nominal pixel size of 0.873 Å. A total of 6689 movies were collected, covering a defocus range of −1.5 to −2.0 µm. The total dose applied was 40 e^−/Å^2, achieved by using a dose rate of approximately 12.5 e^-/s/phys pixel per frame across 40 frames, resulting in a total exposure time of 2.5 s.

### Image processing and 3D reconstruction

The mGlyR cryo-EM dataset was processed using[56] and cryoSPARC[57]. The 6689 movies were initially motion-corrected for beam-induced motion using MotionCor2[58] within RELION. Motion corrected images were then imported into cryoSPARC[57], where the patch Contrast transfer function (CTF) estimation tool was used for CTF estimation.

For particle picking, the TOPAZ algorithm, which employs a convolutional neural network algorithm implemented in CryoSparc, was used[59]. A training set of 1000 micrographs were used to generate a trained model, which was subsequently used for particle picking across the entire dataset, resulting in a total of 3,247,619 particles, which were then extracted from the micrographs with a box size of 340. The extracted particles underwent three rounds of reference-free 2D classification to discard poor-quality particles. The protein particles exhibiting favorable 2D class averages were combined and subjected to several rounds of ab initio and heterogeneous refinement. This iterative refinement process led to the generation of two subsetsone containing 91,859 particles corresponding to the Nb20-mGlyR-RGS complex, and the other consisting of 121,523 particles corresponding to the mGlyR-Nb20 complex. The final resolution of these subsets was estimated to be approximately 3.49 Å and 3.89 Å, respectively, using the gold standard Fourier shell correlation (GSFSC) method within cryoSPARC[57]. To improve the maps for ECD-Nb20 and TM-RGS7-Gβ5, focused refinement was performed using soft masks (0.001 threshold, 12-pixel padding, 3-pixel dilation) corresponding to each region. This focused refinement approach produced maps at 4.14 Å for ECD-Nb20 and 4.18 Å for TM-RGS7-Gβ5, enabling model building. EM density visualization was done in UCSF Chimera[60].

## Model building and refinement

The models of mGlyR-Nb20 and Nb20-mGlyR-RGS7-Gβ5 were built based on the previously determined structure of mGlyR (PDB ID: 7SHE and 7SHF) using COOT[61]. When building the model for mGlyR, the density corresponding to Nb20 was observed, indicating the presence of the bound nanobody. The shape and size of this density were consistent with the bound nanobody. The Nb20 model generated using the AlphaFold method was docked into the cryo-EM map and manually adjusted, built, and refined using COOT[61].

The cryo-EM map of Nb20-mGlyR-RGS7-Gβ5 had a final resolution of 3.89 Å. The map exhibited well-resolved density for the transmembrane (TM) domain but relatively low density for RGS7, with the RGS domain missing from the maps. To build the atomic model of the Nb20-mGlyR-RGS7-Gβ5 structure, the mGlyR-Nb20 model generated in this study (from the same dataset) was used as a template. The model was docked into the map using UCSF Chimera and further manually adjusted, built, and refined in COOT.

In the Nb20-mGlyR-RGS7-Gβ5 map, the density for the RGS domain of RGS7 was poorly resolved and fragmented, making it challenging to build a reliable model. The TM domain of both the mGlyR-Nb20 and Nb20-mGlyR-RGS7-Gβ5 models underwent iterative manual building in COOT, followed by real-space refinement in PHENIX[62]. The Gβ5 and RGS7 regions were not resolved well, and therefore, these regions were docked into the map as rigid bodies. The RGS domain, which is not visible in the map, was not built and remains absent from the final model. Local rotamer fitting and restrained group ADP refinement were also performed. The resulting models were refined against both the unfiltered half maps and sum maps in real space using PHENIX. Structures were visualized, and figures were prepared in UCSF Chimera[60], ChimeraX[63,64] and PyMOL[65]. The data collection and refinement statistics are listed in Table 1.

## Culture and transfection of mammalian cell lines

HEK293FT cells were obtained from ThermoFisher and grown in Dulbecco's modified Eagle's medium (DMEM) supplemented with 10% fetal bovine serum (v/v), minimum Eagle's medium non-essential amino acids, 1 mM sodium pyruvate, and antibiotics (100 units/ml penicillin and 100 mg/ml streptomycin) at 37 °C in a humidified incubator containing 5% CO2. Cells were transiently transfected using Metafectene Pro (Biontex, Germany) in 96 96-well plate following the manufacturer's instructions.

## Cell-based bioluminescence resonance energy transfer (BRET) assays

HEK293FT cells were seeded in a white flat-bottom 96-well plate (Greiner Bio-One) at 50,000 cells per well and transiently transfected using Metafectene Pro with the manufacturer's instructions. pcDNA3.1 plasmids coding for Dopamine D2R (1), mGlyR (1), RGS7 (1), Gb5 (1), Gao (2), Venus-1-155-Gγ2 (1), Venus 156-239-Gβ1 (1) and masGRK3ct-Nluc (1) were used to transfect cells (ratio in parenthesis). An empty vector (pcDNA3.1(+)) was used in order to normalize the quantity of transfected DNA. 24 h after transfection in DMEM complete medium supplemented with 0.1% of Matrigel (Corning), cells were washed with BRET buffer (Dulbecco's Phosphate-Buffered Saline (PBS) containing 0.5 mM MgCl2 and 0.1% glucose). Measurements of BRET between Venus-Gβ1γ2 and masGRK3ct-Nluc were performed to monitor the release of free Gβγ dimers after activation of heterotrimers containing Gα subunits in living cells, as described before[66]. Cells were incubated with 1 μM of Nb20 or with 100 μM of glycine or both when indicated and with the NanoLuc (Nluc) substrate at the manufacturer's instructions. In order to release Gαo and start deactivation of Gαo, 100 μM of dopamine and 100 μM of haloperidol were injected sequentially and automatically (t = 10 s and t = 25 s) by the plate reader (PHERAstar FSX, BMG Labtech), and dual-luminescence was measured at 475 ± 30 nm and 535 ± 30 nm. The

**Table 1 | Cryo-EM data collection, refinement and validation statistics**

| | Nb20-mGlyR (EMDB-65225) (PDB 9VOR) | Nb20-mGlyR-RGS7-Gβ5 (EMDB-65226) (PDB 9VOS) |
|---|---|---|
| **Data collection and processing** | | |
| Voltage (kV) | 300 | 300 |
| Electron exposure (e–/Å²) | 40 | 40 |
| Defocus range (μm) | -1.5 to -2.0 | −1.5 to −2.0 |
| Pixel size (Å) | 0.873 | 0.873 |
| Symmetry imposed | C1 | C1 |
| Initial particle images (no.) | 3,247,619 | 3,247,619 |
| Final particle images (no.) | 121,523 | 91,859 |
| Map resolution (Å) | 3.49 | 3.89 |
| FSC threshold | | |
| Map resolution range (Å) | 2.6 to 7.0 | 2.6 to 7.0 |
| **Refinement** | | |
| Initial model used (PDB code) | 7SHE | 7SHF |
| Model resolution (Å) | 3.49 | 3.89 |
| FSC threshold | | |
| Model resolution range (Å) | 2.6 to 7.0 | 2.6 to 7.0 |
| Map sharpening B factor (Å²) | −159 | −172 |
| **Model composition** | | |
| Non-hydrogen atoms | 10931 | 11,009 |
| Protein residues | 1331 | 1458 |
| Ligands | 24 | 24 |
| **B factors (Å²)** | | |
| Protein | 121.72 | 136.77 |
| Ligand | 140.54 | 150.33 |
| **R.m.s. deviations** | | |
| Bond lengths (Å) | 0.004 | 0.003 |
| Bond angles (°) | 1.5 | 0.721 |
| **Validation** | | |
| MolProbity score | 2.36 | 2.23 |
| Clashscore | 10.7 | 10.0 |
| Poor rotamers (%) | 0 | 0 |
| **Ramachandran plot** | | |
| Favored (%) | 82.73 | 84.30 |
| Allowed (%) | 16.58 | 16.54 |
| Disallowed (%) | 0.69 | 0.94 |

BRET signal was determined by calculating the ratio of the light emitted by the Venus- Gβ1γ2 (535 nm with a 30 nm band path width) over the light emitted by the masGRK3ct-Nluc (475 nm with a 30 nm band path width). The average baseline value (basal BRET ratio) recorded prior to agonist stimulation was subtracted from the experimental BRET signal values, and the resulting difference (net-BRET ratio) was normalized against the maximal netBRET value recorded upon agonist stimulation. The rate constants (1/τ) of the deactivation phases were obtained by fitting a one-phase exponential decay curve to the traces with Graphpad Prism 9.0. The $k_{GAP}$ rate constants were determined by subtracting the basal deactivation rate ($k_{app}$) from the deactivation rate measured in the presence of exogenous RGS protein.

## Flow cytometry

HEK293FT cells were cultured in 6-well plates at a density of $1.10^6$ per well and transfected with 2 μg of cDNA of mGlyR or empty pcDNA3.1+ in control experiments, using Metafectene Pro. 48 h after transfection, cells were mechanically detached by pipetting up and down, washed in PBS supplemented with 0.1% BSA, counted and incubated in PBS-0.1% BSA for 1 h at 4 °C under rotation. Nanobody-20 (Nb20) and 10 μl of anti-myc-APC conjugated antibody (R&d Systems #IC3696A) were added and incubated in the dark with rotation, at 4 °C for 1 h. After 3 washes, cells were analyzed in the LSR-II BD flow cytometer. A gating strategy to sort individual cells from debris and doublets was used as indicated in each figure. Sorted cells were measured for fluorescence in respective channels. Negative control conditions were used to set the positive threshold for APC (mock cells incubated with Nb and anti-myc-APC antibody), and for Venus (mock transfected cells). Acquired data were analyzed using FlowJo software (FlowJo).

## Surface plasmon resonance (SPR)

Surface plasmon resonance (SPR) measurements were performed on a Biacore X100 instrument at 25 °C using 1x HBS-EP+ (Cytiva) as a running buffer. A mouse anti-human IgG CH2 monoclonal antibody (Cytiva) was immobilized to a density of -9500 response units (RU) on a CM5 sensor chip via standard NHS/EDC coupling methods (Cytiva). Subsequently, GPR158-Fc at 10 μg/mL was captured to -1,800 RU on the active flow cell. A concentration series with two-fold dilutions (500–31.25 nM) of the nanobody Nb20 were injected using a multi-cycle method. The lowest concentration (31.25 nM) was repeated to confirm regeneration of the sensor chip. Biacore X100 Control Software 2.0.1 (Cytiva) was used to collect data, and Biacore X100 Evaluation Software 2.0.1 (Cytiva) was used to analyze data.

## Brain slice preparation and whole cell recordings

Electrophysiological recordings from layer II-III neurons of the pre-limbic cortex were performed with mice of either sex aged between 4–12 weeks. Mice were anesthetized with isoflurane and decapitated. The brain was quickly removed and rested for 30 s in ice-cold oxygenated solution containing (in mM): 93 NMDG, 2.5 KCl, 1.2 NaH2PO4, 30 NaHCO3, 20 HEPES, 25 glucose, 2 thiourea, 5 Na-ascorbate, 3 Na-pyruvate, 0.5 CaCl2·, 10 MgCl2, (adjusted to 7.3–7.4 pH with HCl). Coronal slices (300 μm thick) were cut on a vibratome (VT1200S, Leica), mounted on a porous membrane and incubated for 12 min at 34 °C in NMDG. Slices were then transferred to a modified HEPES ACSF containing (in mM): 92 NaCl, 2.5 KCl, 2 CaCl2, 2 MgCl2, 1.2 NaH2PO4, 30 NaHCO3, 20 HEPES, 25 glucose, 5 Na-ascorbate, 2 thiourea, 3 Na-pyruvate, (adjusted to 7.3–7.4 pH with NaOH) and allowed to recover for 1 h at room temperature. For recordings, slices were transferred to a submerged recording chamber where they were continuously perfused at 2 ml/min with oxygenated ACSF containing the following (in mM): 126 NaCl, 2.5 KCl, 2 CaCl2, 2 MgCl2, 18 NaHCO3, 1.2 NaH2PO4, 10 glucose, in presence of the following synaptic blockers: picrotoxin (100 μM), strychnine (1 μM), CNQX (20 μM), APV (50 μM). Pipets (3–5 MΩ) were pulled from P-1000 (Sutter Instruments, CA) and filled with an intracellular solution containing the following (in mM): 119 K-MeSO4, 12 KCl, 1 MgCl2, 0.1 CaCl2, 10 HEPES, 1 EGTA, 0.4 Na-GTP, 2 Mg-ATP (280–300 mOsm, pH 7.3 adjusted with KOH). Slices were incubated with Nb-20, Nb-20* (1 μM), or ACSF and changes in neuronal excitability were assessed by counting the number of spikes evoked in response to a 1-s depolarizing ramp ranging from 0 to 200 pA with a 20-s intertrial interval. The Rheobase current was defined as the minimum current necessary to elicit the first AP. Acquisition was done using Clampex 10.7, MultiClamp 700B amplifier and Digidata 1440 A (Molecular Devices, CA). Data were analyzed with Clampfit 10.7.

## Mouse studies

No statistical method was used to predetermine sample size. No data were excluded from the analyses. No specific randomization methods were used. Animals were randomly assigned to experimental groups depending on genotype. Experimenters were blinded to the treatment groups.

**Drug treatments.** Intracerebroventricular (ICV) administration was performed according to a previously described method[67]. Mice were injected with 5 μl of purified and endotoxin-free nanobody solution in saline (1.92 mg/ml) of nanobody or 5 μl of vehicle. Behavioral evaluation was conducted starting 24 h after treatment over the course of three days. Marble burying was performed in the morning on day one, followed by the elevated plus maze in the evening. Tail suspension was done the next day, and the force swim test was performed on day three. The same sequence and timing of behavioral testing was repeated two weeks after treatment. For the intranasal (IN) administration, 24 h after the last stressor, mice received two treatments on the same day, separated by a 6-h interval. Each treatment involved the application of 10 μl of purified, endotoxin-free nanobody solution in saline (0.95 mg/ml) or racemic ketamine hydrochloride (20 mg/kg; VetaKet; Patterson Veterinary; Cat# 78925834) dissolved in saline, delivering 5 uL to each nostril, using a P-10 pipette.

**Stress paradigm.** The chronic variable stress (CVS) paradigm consists of daily exposure to one of three stressors over 21 consecutive days. All mice are exposed to one stressor per day, and the three stressors are repeated every three days. The times for each stressor vary from day to day to increase unpredictability. The stressors include restraint stress, where mice are placed in ventilated 50 mL conical tubes for 1 h in the home cage. The next stressor is foot shock, which is performed in shock boxes (6 mice can be simultaneously run). Mice receive 100 mild foot shocks at 0.4 mA for 1 h at random intervals. The final stressor is 30 min of predator odor exposure. 15 uL of TMT (Fisher Scientific; 501844430) is pipetted onto a cotton tip applicator and placed into a clean, empty standard mouse cage. The single housing is also considered an additional stressor on the mice. Termination criteria were used to determine whether mice should continue in the stress paradigm, including bleeding, excessive weight loss (>20% initial), and hunched or moribund phenotypes. Behavior was run 24 h after drug treatment in counterbalanced groups, and animals were numbered to maintain blinding in the manually scored tests. 48 h after the final behavioral test, mice were sacrificed.

**Marble burying.** Marble burying (MB) was conducted in a standard mouse cage (27 × 16.5 × 12.5 cm) with 5 cm corncob bedding and 20 glass marbles overlaid in a 4 × 5 equidistant arrangement. Background white noise (approximately 70 dB) was used during trials. The mouse was placed in the center of the cage, and testing consisted of a 30 min exploration period. The marbles, at least half-buried at the end of the trial, were counted as buried.

**Sucrose splash test.** The splash test (ST) was performed using a 10% sucrose solution freshly prepared the day of the test (Sigma-Aldrich; S9378) and was sprayed onto the dorsal coat of the mouse (-0.35 mL per mouse), and placed into an empty, inescapable cylindrical Plexiglas container (121 cm in length and 15 cm in diameter). Mice were habituated to the cylinder for 5 min before spraying with sucrose solution and returning to the container. The behavior of the mouse was recorded for 5 min and later scored manually by blinded observers. The videos were scored for total grooming time. The scores of the observers are averaged to give final values.

**Elevated plus maze.** The elevated plus maze (EPM) was performed using a black, plexiglass elevated plus maze (apparatus with two open

and two enclosed arms, 33 × 6 cm, with a wall of 25 cm on the closed arm, elevated 60 cm from the floor; Med Associates, St. Albans, VT). Lighting for the maze was set at 200 lux in the center of the plus maze, 270 lux on the open arms, and 120 lux on the closed arms. Background white noise (approximately 70 dB) was used during trials. Mice were placed in the center of the elevated plus maze and left to explore for 5 min in a dim light condition. Mice were recorded using Ethovision XT, and the time spent in the open and closed arms, as well as the number of entries from the closed to the open arm, were calculated.

**Tail suspension test**. The tails of the mice were wrapped with tape that covered approximately 4/5 of the tail length and then fixed upside down on a hook. The immobility time of each mouse was recorded and tracked over a 6 min period using Ethovision XT. Automated tracking values were validated via manual scoring by a blind observer.

**Force swim test**. The Porsolt Forced Swim Test (FST) was conducted using a vertical clear glass cylinder (10 cm in diameter, 25 cm in height) filled with water (25 °C). The mice spent 6 min in the water, and immobility was scored. A mouse was regarded as immobile when floating motionless or making only those movements necessary to keep its head above the water. Automated tracking values were validated via manual scoring by a blind observer.

**Analysis of emotionality score**. The behavioral paradigm used to calculate the emotionality score were performed in the following order: MB, ST, EPM, TST and FST. To obtain a comprehensive measure for emotionality, we used z-scoring methodology to integrate standard measures of anxiety-like and depressive-like behaviors, as previously described[68]. The testing parameters analyzed were as follows: marble burying (number of marbles buried), elevated plus maze (time spent on open arm, number of entries into the open arm), tail suspension test (immobility) and forced swim test (immobility). For each parameter, the z-score for every individual animal was calculated using the previously described approach[20]. Briefly, for each parameter, the z-score of each individual animal was calculated using the formula $Z = \frac{X - \mu}{\sigma}$, where X represents the individual data point, m represents the mean of the control group, and s represents the standard deviation of the control group. The emotionality score (ES) for each individual subject was first averaged within test, and then across each test to ensure equal weighting of all tests. $ES = \frac{Z_{MB} + Z_{EPM} + Z_{TST} + Z_{FST}}{number\ of\ tests}$. The mean emotionality score for each group is an average of the individual scores within each group for each experiment.

### Novelty-suppressed feeding

Novelty-suppressed feeding (NSF) was measured in 24-h food-deprived mice, isolated in a standard housing cage for 30 min before individual testing. Three pellets of ordinary lab chow were placed on a white tissue in the center of each arena, lit at 80 lux. Each mouse was placed in a corner of an arena and allowed to explore for a maximum of 15 min. Latency to feed was measured as the time necessary to bite a food pellet. Immediately after an eating event, the mouse was transferred back to the home cage (free from cage-mates) and allowed to feed on lab chow for 5 min. Food consumption in the home cage was measured.

### Novel object recognition

The experiments were conducted in 4 equal square arenas (50 × 50 cm) separated by 35 cm-high opaque gray Plexiglas walls. The light intensity of the room was set at 15 lux to facilitate exploration and minimize anxiety levels. The floor was a white Plexiglas platform (View Point, Lyon, France), spread with sawdust. The room was equipped with an overhead video camera connected to a computerized interface, allowing visualization and recording of behavioral sessions on a computer screen in the adjacent room. The experimental paradigm

lasted for 2 days (3, 4). On day 1, the animals were placed in an arena for a 15 min-habituation session with two copies of an unfamiliar object (T-shaped plastic tubing, 1.5 × 3.5 cm). These objects were not used later for the recognition test. On day 2, the recognition test was performed, consisting of 3 trials of 10 min separated by 2 intertrial intervals of 5 min, during which the animals were returned to their home cage. On the first trial, or familiarization phase, the mice were presented with two copies of an unfamiliar object. On the second trial, or place phase, one of the two copies was displaced to a novel location in the arena. On the third trial, or object phase, the copy that had not been moved on the previous trial was replaced by a novel object. Stimuli objects used in all previous experiments were Lego bricks, plastic rings, dice or marbles (size 1.5–3 × 2–3 cm). The identity of the objects, as well as the spatial location in which these objects were positioned, was balanced between subjects. The number of visits and the time spent exploring each object were scored manually on video recordings. A visit was acknowledged when the nose of the mouse came in direct contact with an object. A percentage of discrimination was calculated for the number of visits and time exploring the objects as following: exploration of displaced or novel objects / total exploration. The discrimination ratio during the familiarization phase was arbitrarily calculated for the object located in the right upper corner of the arena.

### Open field

Locomotor activity was monitored at 15 lux either in open fields placed over a white Plexiglas infrared-lit platform. Locomotor activity was recorded via an automated tracking system (Ethovision, Noldus, Wageningen, Netherlands).

### Rotarod

The apparatus (UgoBasile, Germonio, Italy) was set to accelerate from 4 to 40 rpm in 5 min. On day 1, mice were habituated to rotation on the rod at 4 rpm, until they were able to stay more than 180 s. On day 2, mice were tested for three daily trials (60 s intertrial). Each trial started by placing the mouse on the rod and beginning rotation at a constant 4 rpm-speed for 60 s. Then the accelerating program was launched, and the trial ended for a particular mouse when falling off the rod. Time stayed on the rod was automatically recorded.

### Blood collection and corticosterone measurements

Mice were euthanized by rapid decapitation 24 h after treatment with Nb20 or vehicle/PBS (48 h after last stressor). Trunk blood was collected in Eppendorf tubes and allowed to clot for 30 min. Serum was isolated after 2000 g centrifugation and stored at −20 °C. Corticosterone levels were measured using a commercially available ELIZA kit (ThermoFisher #EIACORT) according to the manufacturer's instructions.

### Immunogenicity analysis

Amino acid sequence of nanobodies was analyzed using NetMHCIIpan version 4.0 to predict binding affinities to MHC class II molecules[69]. The sequence was submitted in FASTA format to the online tool, and binding predictions were generated for a comprehensive panel of MHC class II alleles. Binding strength was categorized based on percentile ranks, with published thresholds identifying strong, weak, and non-binders.

### Data analysis and statistics

The functional data shown represent the average ± SEM of at least 3 individual experiments each performed in triplicate and expressed as means ± standard error means (s.e.m). For the GAP assay, cells with the same transfection conditions were compared in their treatment and a two-way ANOVA test was performed. The non-linear regression curve was used to fit the dose-response curve, giving the IC$_{50}$ value. For the flow cytometry binding assays, cells transfected with an empty vector

were used as a control, and the non-linear regression curve was used to fit the dose response curve, giving the $EC_{50}$ value. For the SPR, calculation of association ($k_{on}$) and dissociation ($k_{off}$) rate constants was based on a 1:1 Langmuir binding model. For the mouse experiments, Marble Burying, Splash Test, Tail Suspension, Elevated Plus Maze and Force Swim Test, comparisons were by one-way ANOVA with Dunnett or Kruskal-Wallis post hoc tests. Assumptions of normality and homogeneity of variance were studied using Shapiro–Wilk tests and Levene's tests, respectively. $*P < 0.05$; $**P < 0.01$; $***P < 0.001$; $****P < 0.0001$. Mean values with s.e.m. are shown. For the ELIZA, a one-way ANOVA test was performed to compare the control treatment with the treated condition. $*P < 0.05$; $**P < 0.01$; $***P < 0.001$; $****P < 0.0001$. Mean values with s.e.m. are shown. For the electrophysiology experiments, changes was assessed in a nonparametric $t$-test; the Wilcoxon test. Values of $*P < 0.05$, $**P < 0.01$ and $***P < 0.001$ $***P < 0.0001$ were considered to be statistically significant. Calculations, graphs and statistics were generated using GraphPad Prism 9 software (San Diego, CA, USA).

## Reporting summary

Further information on research design is available in the Nature Portfolio Reporting Summary linked to this article.

## Data availability

The atomic coordinates have been deposited in the Protein Data Bank (PDB) under accession codes 9VOR and 9VOS. Single-particle cryo-EM maps are available at the Electron Microscopy Data Bank (EMDB): EMD-65225 and EMD-65226. Previously published atomic coordinates used in this study are 7SHF and 7EWP. The source data underlying Fig. 1, Fig. 2, Fig. 4, Fig. 5 and Supplementary Fig. S7 and Supplementary Fig. 9–18 are provided as a Source Data file. The data generated in this study have been deposited in the Figshare [https://doi.org/10.6084/m9.figshare.30714752]. Source data are provided with this paper.

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

## Acknowledgments

This work was supported by NIH Grants MH105482 (to K.A.M.), EY034339 (to K.A.M. and A.K.S.), IC-12048(12)/1/2022-ICD-DBT (to A.K.S), ICMR (5/4-5/3/36/Neuro/2022-NCD-I) (to A.K.S.) J.A.J.B and J. L. M. acknowledge funding from Région Center Val de Loire (ARD2020 Biomédicament – GPCRAb). Their work was supported by the Institut National de la Santé et de la Recherche Médicale (Inserm), Center National de la Recherche Scientifique (CNRS), Institut National de Recherche pour l'Agriculture, l'Alimentation et l'Environnement (INRAE) and Université de Tours. K.V.N. acknowledges the MHRD, and M.S. acknowledges the CSIR, India, for the fellowships. This research was, in part, supported by the National Cancer Institute's National CryoEM Facility at the Frederick National Laboratory for Cancer Research under contract HSSN261200800001E. We thank Dr. Dipak Patil for purifying proteins, preparation of samples for the CryoEM and help with data acquisition, Ms. Natalia Martemyanova for help with mouse husbandry, Ms. Dao for technical help with mouse behavior experiments and members of the Martemyanov Lab for helpful discussions. We thank the staff, Adam Wier, Thomas Edwards and Ulrich Baxa of the NCI National CryoEM Facility for data collection. We also thank Dabbu Kumar Jaijyan for help with the cryo-EM sample. We thank Servier Medical Art (http://www.servier.com) for providing templates for graphics, which were adapted into cartoons.

## Author contributions

T.L. and K.A.M. conceived the project; T.L. performed all cell signaling, flow cytometry and biochemical experiments and analyzed behavioral data. S.Z. performed electrophysiological experiments, O.K.S. and G.B. performed mouse behavior, S.S. prepared samples for CryoEM and acquired data, M.S., S.S., K.V.N., and A.K.S. analyzed CryoEM data and solved the structure, H.P. and C.R. performed biochemical experiments characterizing Nb20 binding by SPR, J.A.J.B. and J. L. M. developed a nanobody library. A.K.S. supervised structural biology and analyzed the structure. K.A.M. supervised the project and co-wrote the paper with T.L. and A.K.S., with feedback from all authors.

## Competing interests

K.A.M. is a co-founder of EvoDenovo Inc., a startup company pursuing mGlyR as a drug target. The remaining authors declare no competing interests.
