## [Transparent Peer Review file · Nature Communications]

Targeting mGlyR with nanobodies for depression

Corresponding Author: Professor Kirill Martemyanov

Version 0:

Reviewer comments:

Reviewer #1

(Remarks to the Author)

In this study, Laboute et al developed a nanobody (Nb20) targeting mGlyR and found that Nb20 functions as an antagonist for mGlyR. Authors performed cell-based analysis, structure determination and animal model experiments to claim that Nb20 binds the Cache domain of mGlyR and down-regulates its GAP activity for the anti-depressant effects. The overall idea and approaches are sounding and authors showed interesting data in cell-based and animal model analyses. The main problems in this work are in structure analysis part, which are described below. Authors should carefully re-analyze their structures to provide convincing links between the structures and their biological data to improve the quality of this manuscript.

1. P6, I11 "...we found that the introduction of RGS7/Gb5 accelerated deactivation of its substrate, Gao (Fig.2B,D). Application of Nb20 had no significant effect on either baseline Gao deactivation or RGS7/Gb5-assisted process (Fig.2B,D)". Fig 2D, it is not clear where the middle RGS7/Gb5 (GPR158 -, RGS7/Gb5 +) data comes from. Unlike those in the left (-/-) and right (+/+) in the panel, the source in the middle is unclear. If it is from 2B, the bar graph cannot be same or similar to those of -/-.

2. Page 6, I -3 "The lack of interplay between Nb20 and glycine likely indicates distinct mechanisms of mGlyR activity modulation by the two ligands and inability of Nb20 to access glycine binding pocket (Supplementary Fig. 4)." In Fig. 4B, Nb20 and glycine do not exhibit a synergistic effect, and there is no available structure of glycine-bound GPR158. Therefore, the proposed distinct mechanisms by which Nb20 and glycine require additional supporting evidence. Furthermore, on p14: I 5, "Our functional experiments reveal no interaction between glycine and Nb20 effects. While this suggests that Nb20 and glycine may share a similar mechanism of mGlyR inhibition, there is currently no structure of glycine-bound mGlyR, and future experiments will need to explore whether Nb20 taps into the mechanism of mGlyR antagonism utilized by its endogenous ligand glycine." This statement is inconsistent with the earlier claim of distinct mechanisms.

3. P8, "We validated the binding mode of Nb20 by mutating the CDR1 and CDR2 regions involved in the binding which resulted in a complete loss of interaction of mutant Nb20 (Nb20*) with mGlyR (Supplementary Fig. 6 & 8)." Authors should provide more detailed information including the mutated residue information to help readers better understand the texts. Additionally, please explain how Supplementary Figure 6 supports the authors claims.

4. P8. Structure superposition (Supp Fig 9D and E). Authors should compare with other available structures and perform more careful analysis. To better demonstrate the independent effect of Nb20 on the GAP activity of the mGlyR-RGS7-Gb5 complex, it would be more appropriate to compare the TM structures between mGlyR-RGS7/Gb5 and Nb20-mGlyR-RGS7/Gb5.

5. The structures are not properly analyzed or described in the text. There are many inconsistencies in the authors' structures and their description in the text. These include residue name, number and interactions, and cannot be considered as typo. I provided some examples here, but authors should carefully check other parts too.

(i) P7, I -2. "... π - π stacking..between W57 and W162; I am not convinced if they form π - π stacking – please see their

orientation and distance.

(ii) P8, l 1. There is no N142; S142 and N143 – in any case, both residues are over 7 Å away from Y60.

(iii) P8, l 3, No D198, E198 is far apart from K79 to form a hydrogen bond.

Also, considering these features, it is not clear how mutations in these residues affect the interaction between Nb20 and mGlyR (Supp Fig 6 and 8).

6. P9, l2, "...In the mGlyR-RGS7/Gβ5 structure, the ECD domain could not be observed.... the binding of Nb20 stabilized both the ECD and RGS7-Gβ5, enabling the reconstruction of the entire assembly with reasonably well-resolved density." This is a critical issue as the sentence is not true. The key feature of the authors' model is that Nb20 binds and stabilizes the extracellular domain, which was not visible in Nb20-free state. Subsequently the conformational change is relayed to the TM and intracellular region, which alters the structure of the RGS7-Gβ5. In contrast to authors claim, the extracellular domain, TM and RGS7-Gβ5 are well defined in the reported Nb20-free mGlyR-RGS7/Gβ5 (PDB 7EWP). Because this is the starting point of the authors' model for the structural basis for the Nb20-mediated anti-depressant effect of mGlyR, the authors' proposed model is not consistent with available structures. I strongly suggest that the authors should reinterpret their data by comparing with all available structures including 7EWP.

7.P9. Even in the structure after focused refinement, the EM density for Nb20 are not well-defined. This is observed from the pdb and map from the authors and Supplementary Fig. 7D. Thus, one possibility is that the end-regions of the complex including Nb20 and Gb5 are flexible instead of the Nb20-binding induced release of RGS7-Gb5 from mGlyR. Authors may consider such possibility. Also, the model-fitted-map seems unmatched each other in supplementary Fig. 7D. It would be helpful that authors provide better figure or check their orientations.

8. P 9. "...Even after local refinement and analysis at various contour levels, the RGS domain density remained unresolved...":

Authors may consider to perform structural analysis using a newly generated local map encompassing the 7TM-RGS-Gβ5 regions. At the current resolution, the map quality is still poor to support claims that the nanobody induced the unresolved Gβ5 region. Please consider increasing the contour level to reduce noise from detergent micelles. If the signal is sufficiently strong, the 7TM-RGS7-Gβ5 complex—excluding the unresolved regions—should remain clearly visible. This would help determine whether the unresolved areas are due to localized disorder, rather than overall weak density across the entire RGS7-Gβ5 region. Then, the authors should represent these maps accordingly. The current maps shown in Supplementary Fig. 7F appear to be at low contour levels.

9. P9. Considering the weak density of intracellular region and more than half of the EM density for Gb5 is missing, it is difficult to say 5 Å translation of the Gb5 in the presence of Nb20. Description of the conformational change with inaccurate structure could mislead the readers and carefully addressed.

10. P9. "Nb20 binding disrupts key hydrogen bonds: one between R715 from CTH2 and E172 from RGS7, and another between T602 from the TM domain and D147 from RGS7 (Supplementary Fig. 9H)." T602 and D147 do not form hydrogen bonds in the 7SHF structure. Authors should check this part. Calculate local resolution and present it as a figure to claim residue-shift. In my opinion, 4 Å resolution is not sufficient to claim the residue shift.

11. The FSC curve for the locally refined map (Supplementary Fig. 6G) shows an abnormally large bump, suggesting overfitting due to an overly tight mask. One of the binary masks shown in Supplementary Fig. 5 also appears too tight, excluding signal near the edge regions. The authors should consider to repeat the local refinement using a new mask generated from a much larger base mask, for example using ChimeraX.

12. Overall resolution is insufficient to claim the loop shifts or residue shifts (Supplementary Fig. 9D, E, H). The authors are highly encouraged to present model-fitted-maps for all regions separately.

Minor comments

1. P7 l9, "To improve the map, local refinements using soft mak corresponding to ECD-Nb20"
Typo, refiments > refinements, soft mak > soft mask.

2. Supplementary Fig. 9C and Supplementary Fig. 9F. Please indicate the the position of the stalk domain.

3. Are Fig. 3C and 3D in the same view? It would be helpful to match their viewpoints for easier comparison. Additionally, indicating the positions of CDR1 and CDR2 would help readers to interpret the figure.

Reviewer #2

(Remarks to the Author)

The revised manuscript has significantly improved and addresses my previous main concerns. The inclusion of additional behavioral tests strengthens the interpretation of Nb20's effects on depression-related phenotypes. Importantly, the authors now provide data on hedonic behavior, sex differences, and corticosterone levels following administration. Clarifications regarding the EPM findings over time and additional behavioral parameters such as arm entries and head-dips have also been included and are clearly presented. I find the revisions acceptable, and I believe the manuscript is now suitable for publication.

Reviewer #3

(Remarks to the Author)

The authors develop a new nanobody-based strategy to antagonize the mGlyR, which rescues behavioral deficits associated with chronic stress and psychiatric disease. The authors show that the Nb20 nanobody selectively modulates mGlyR function and defines the downstream biochemical and structural mechanisms, as well as the electrophysiological consequences. Multiple delivery approaches of Nb20 induce antidepressive-like and long-lasting phenotypes in a variety of behavioral assays. These behavioral effects are associated with increased excitability of neurons in the prelimbic cortex, where mGlyR is highly expressed. This work reflects a new and exciting angle to target brain-specific antigens, with intranasal delivery of a nanobody highlighting translational potential. I have a few suggestions that would strengthen the central claims of the manuscript.

1.) Is the Nb20 antibody specific to mGlyR *in vivo*? The null nanobody controls for off-target effects of the surgery and nanobody domain, but does not rule out promiscuous binding of Nb20 to other targets, which could generate indirect effects on behavior. Since the authors studied mGlyR knockout mice, are the antidepressant-like behavioral effects of Nb20 affected in mice lacking the expected antigen?

2.) In Fig 5, the authors confirm that Nb20-based mGlyR antagonism phenocopies the increased excitability of L2/3 PFC neurons in mGlyR KO mice, which is a valuable control. However, it's not clear how this relates to the behavioral effects observed after Nb20 treatment in Fig 4 since the treatment timescales differ significantly (10 min treatment *in vitro* in Fig 5 vs 24+ hrs *in vivo* in Fig 4) and these effects may be correlative. If the authors performed these electrophysiology experiments from mice treated with Nb20 (as in Fig 4), are these neurons more excitable? Alternatively, are proxies of excitability, such as cFos, increased in mice treated with Nb20?

3.) While the results are indeed exciting, the manuscript appears overly hyperbolic in several areas. For example, in the discussion they "report a unique immunotherapy solution for a major neuropsychiatric condition- major depressive disorder", but depression reflects a human condition with environmental drivers that are not addressed in this manuscript. Additionally, the authors claim immunotherapies offer "low toxicity", but one of the current limitations of immunotherapies is toxicity, partially due to an incomplete mechanistic understanding (which is highlighted by points 1 and 2 above).

Version 1:

Reviewer comments:

Reviewer #1

(Remarks to the Author)

In the revised text, although authors have provided some answers on questions raised and improved the manuscript, several important issues are remained in unresolved states and these must be properly responded. Also, there are several incorrect descriptions in the text that should be carefully examined and corrected with proper citations.

1.. Supple Fig 6 a, c, d, f are still messy and too complicated.

(i) Authors should provide better views or sections of the maps and models. Fig S6f. Even at very low contour level (near background) of the map where micelle completely covers the TMs, very poor density for RGS7 is shown. I do not think it is proper to build the model here. Accordingly, description of the conformation changes in RGS7 is unlikely to be accurate (as density for the secondary structures for RGS7 is unclear) and should be removed.

(ii) Also, in FigS6f, some TMs do not form helices but rather they form loops (see figure S6f).

(iii) Method section: Authors described "...manual building...". How RGS7 – Gb5 can be manually built? Authors should more precisely describe what they have done. Also, authors should describe which parts of the models are visible in the maps either in the main text or methods section. In principle, model should not fit in the invisible region.

2. P7, line 2 and supple Fig 4 & 5,

(i) the resolution is unlikely to be 3.83 Å. In S5D, the GSFSC curve decreases and increases again near 0.143 (unusual type of curve), and this is unlikely to be the true resolution.

(ii) Authors wrote 3.83 Å in the text, but 3.89 Å in the figure section – which one is correct? Also 3.43 vs 3.49 ? in S5D.

(iii) Also, authors should provide the local resolution maps for the structure with different colors. Possibly, they may represent maps including local resolution at the bottom of Fig S4.

3. The focused refined EM density for the Nb20-ECD is not sufficient to fit the side chains in most of the regions. Therefore, description of the interface interaction could provide wrong information. Some examples include;

(i) E198 – N31; shorter one has 4.5 Å distance, the other one is over 9 Å and both are too far. Moreover, carbonyl - carboxyl oxygen in shorter one

(ii) between R56 of Nb20 and E166 of subunit A: density for the R56 side-chain is not visible unless you lower the contour level near the background level.

(iii) Nb20* mutant: the mutation study performed here is more likely complete disruption of the local core structure CDR1 and CDR2 rather than disruption of the interaction by the surface mutation. I do appreciate if authors have attempted to provide clearer mutational effects, but at least brief comment would help.

Also, in method section: Nb20*: residue number "29 to 35" vs "30 – 35" in the maintext & others. These numbers are

inconsistent number with PDB, which are the correct numbers?

4. P 8 “.. mGlyR– RGS7–Gβ5 (PDB: 7SHF, 7EWP) with that of the Nb20-bound complex. While the overall TM domains are largely conserved, notable deviations of up to 5.5 Å were observed in TM3, TM4, and the connecting loop (Supplementary Fig. 8F, H)...”

I have compared the TMs (TM3, TM4 and other TMs) and linkers between the current model (mGlyR-Nb20-RGS-Focused2.pdb) and 7EWP by aligning entire TMDs, and found out no notable differences. In fact, the largest differences in TM3, TM4 and connecting loops are near or less than 1 Å. Authors made inaccurate description and must correct or delete the sentence.

5. “that this domain becomes highly flexible upon binding the Nb20 to ECD. Superimposition of Nb20-bound and free structures by the 7TM region suggested plausible changes.....that remodeling at the 7TM interface triggers changes in contacting residues of...”

Since there are no noticeable changes in TM, this sentence is not correct. Considering TMD retains same structure upon Nb20 binding, authors should reconsider their proposed model.

6. ““In the mGlyR-RGS7/Gβ5 structures, the ECD domain was highly flexible, resulting in low resolution features as observed in 7EWP or scattered density at relatively higher resolution in 7SHF structures. The binding of Nb20 appeared to stabilize both the ECD and RGS7-Gβ5 module, yielding relatively well-resolved density for ECD. Strikingly, comparison revealed that the density corresponding to the RGS domain, which typically wraps around Gβ5 from below, was poorly resolved in the Nb20-bound structure, suggesting that RGS domain adopts””

This is not properly revised paragraph. This is only to justify their claims that Nb20 stabilizes ECD. Resolution of the 7EWP ECD is not sufficient to place the side-chain, but main-chain for ECD is well defined. Nb20-bound EM map is no better (to place the side-chain) than the Nb-free map (7EWP) to claim that Nb20 stabilizes ECD of mGlyR.

(i) low resolution vs high resolution: Description of resolution is meaningful only if the features are present and can be comparable. This sentence gives impression that the ECD of 7EWP is lower resolution than that of 7SHF. However, 7SHF does not have ECD, a key feature authors described. Comparison of the local resolution of ECDs between 7EWP and 7SHF would be reasonable. Thus, there is no point of describing resolution here, and should be removed as this type of description only confuses the readers.

(ii) For the readers to clearly understand, the references for 7EWP and 7SHF should be included.

(iii) After careful checking of the cryo-EM map for the ECD in Nb20-free and Nb20-bound, I cannot agree with authors claims that Nb20-binding stabilizes ECD and their further idea.

7. “..but relatively low resolution in the ECD and Nb20 binding regions. Nevertheless, these maps allowed us to construct the complete models of the mGlyR-Nb20 and mGlyR-RGS complexes,”.

The map for the RGS complex is poorly defined and not sufficient to build the complete model. As I commented above, it is not possible to trace the main-chains for RGS7 and Gβ5. Thus, authors should revise this part.

8. Table 1 for the statistics for the structural analysis. Why only two PDBs are reported? Are authors not going to deposit the focused-refined structures? Also, authors should clearly describe which data will be available to public with their statistics.

Reviewer #3

(Remarks to the Author)

Although my concerns regarding nanobody effects on neuronal function related to the observed behavioral effects were not directly addressed experimentally, given the scope and focus of the study, I acknowledge that the authors have appropriately discussed these limitations in the revised manuscript. I have no further comments or suggestions.

Version 2:

Reviewer comments:

Reviewer #1

(Remarks to the Author)

To explain the Nb20's anti-depression effect, authors proposed a model: Binding of Nb20 to the extracellular ligand binding domain of mGlyR promotes specific stabilized conformation of the receptor which propagates through the stalk region to remodel the 7TM region, which then serve as the contact site for RGS7. Authors provided a detailed description in the text; “...remodeling at the 7TM interface triggers changes in contacting residues of the RGS complex. Nb20 binding appeared to shift CTH2 by 2.2 Å, induce a 1.2 Å displacement in RGS7, and 1.5 Å movement in the TM3-TM4 connecting loop, and up to a 5.5 Å (when compared with 7SHF) changes in the TM3, which serve as the contact site for RGS7...”

In previous review, I have commented that comparison of the structures of the 7TMs between the Nb20-bound- and Nb20-free mGlyR-RGS7/Gβ5 (PDB ID: 7EWP) showed less than 1 Å differences in Ca atoms. While authors agreed with my observation on small structural difference in 7TMs, they maintained their claim that the 7TMs has been remodeled by Nb20-binding, and retained their description in P9 in the main-text. I am not convinced on authors' claim. I understand that the authors tried to explain structural differences using their previously published structure alone (PDB ID: 7SHF). But there is other structure available which should be also used for comparison. This is very simple issue. There is Nb20-free mGlyR-

RGS7/G β 5 structure which is determined independently and has been available in database for several years. Comparison with this structure with the current Nb20-bound mGlyR is unbiased. Its 7TMs structure is virtually identical to that of the Nb20-bound mGlyR-RGS7/G β 5 presented in this work. Can authors claim structural changes in 7TMs, although Nb20-binding does not affect the structure of 7TMs of mGlyR? If so, how would the authors explain the structural identity/similarity of 7TMs between them? Instead of maintaining their claims on conformational changes in 7TMs, authors may consider other possibilities. In my opinion, authors can use Nb20-mGlyR interaction, their strong biological data with animal model, which would make more clear story.

We appreciate continuing interest of the reviewers in this work and many helpful suggestions. We have incorporated all of feedback from the reviewers in the revised manuscript and are grateful for reviewer's time and help in improving this manuscript. Our responses to specific reviewer's comments are as follows. Changes in the manuscript are highlighted in blue font.

Reviewer #1

Comment 1. P6, 111 "...we found that the introduction of RGS7/Gb5 accelerated deactivation of its substrate, Gao (Fig.2B,D). Application of Nb20 had no significant effect on either baseline Gao deactivation or RGS7/Gb5-assisted process (Fig.2B,D)".

Fig 2D, it is not clear where the middle RGS7/G β 5 (GPR158 -, RGS7/G β 5 +) data comes from. Unlike those in the left (-/-) and right (+/+) in the panel, the source in the middle is unclear. If it is from 2B, the bar graph cannot be same or similar to those of -/-.

Response: We apologize for not making presentation of these data consistent. In fact, we collected the entire dataset with all conditions (baseline, No RGS, No mGlyR) with or without Nb20 in parallel. However, the Figure as presented showed a subset of this data as representative traces in panels B and C and another subset as quantification in panel D. We agree that this is confusing. To rectify it, we are now showing the entire dataset aligning all the conditions between panels B, C and D. Thank you for bringing it to our attention.

Comment 2. Page 6, 1 -3 "The lack of interplay between Nb20 and glycine likely indicates distinct mechanisms of mGlyR activity modulation by the two ligands and inability of Nb20 to access glycine binding pocket (Supplementary Fig. 4)."

In Fig. 4B, Nb20 and glycine do not exhibit a synergistic effect, and there is no available structure of glycine-bound GPR158. Therefore, the proposed distinct mechanisms by which Nb20 and glycine require additional supporting evidence.

Furthermore, on p14: 15, "Our functional experiments reveal no interaction between glycine and Nb20 effects. While this suggests that Nb20 and glycine may share a similar mechanism of mGlyR inhibition, there is currently no structure of glycine-bound mGlyR, and future experiments will need to explore whether Nb20 taps into the mechanism of mGlyR antagonism utilized by its endogenous ligand glycine."

This statement is inconsistent with the earlier claim of distinct mechanisms.

Response: We agree with the reviewer that our treatment of Nb20 vs glycine effects have been confusing. The difficulty is largely explained by the lack of the glycine bound structure of mGlyR. We feel that in the absence of such structure any claims regarding the similarity or differences between Nb20 and glycine in terms of the mechanism of action will be highly speculative. With this, we decided to remove the preliminary data on the combinatorial effects of Nb20 and glycine reserving thorough investigation on the mechanisms to the future when glycine bound structure of mGlyR becomes available and/or we interrogate their interaction with focused biophysical and biochemical approaches which seem to fall outside of the focus of this study.

Comment 3. P8, “We validated the binding mode of Nb20 by mutating the CDR1 and CDR2 regions involved in the binding which resulted in a complete loss of interaction of mutant Nb20 (Nb20*) with mGlyR (Supplementary Fig. 6 & 8).”

Authors should provide more detailed information including the mutated residue information to help readers better understand the texts. Additionally, please explain how Supplementary Figure 6 supports the authors claims.

Response: The information regarding the exact residues mutated to generate control NB20* has been provided in the Methods section. We have now revised the main text to provide better description and reference the Methods section. We apologize for inadvertently calling out Supplementary Fig 8 in this section – it is indeed irrelevant to the point regarding Nb20* design or performance.

4. P8. Structure superposition (Supp Fig 9D and E). Authors should compare with other available structures and perform more careful analysis. To better demonstrate the independent effect of Nb20 on the GAP activity of the mGlyR-RGS7-G β 5 complex, it would be more appropriate to compare the TM structures between mGlyR-RGS7/G β 5 and Nb20-mGlyR-RGS7/G β 5.

Response: Thank you for this valuable suggestion. We have now performed a transmembrane (TM) region-based structural comparison between the mGlyR–RGS7/G β 5 complex and the Nb20-bound mGlyR–RGS7/G β 5 complex, as well as with other available related structures (PDB:7EWP). The revised text and Supplementary Fig. 9F, H (Now Supplementary Figure 8) have been updated accordingly (page 8).

5. The structures are not properly analyzed or described in the text. There are many inconsistencies in the authors’ structures and their description in the text. These include residue name, number and interactions, and cannot be considered as typo. I provided some examples here, but authors should carefully check other parts too.

(i) P7, 1 -2. “ .. π - π stacking..between W57 and W162; I am not convinced if they form π - π stacking – please see their orientation and distance

Response: Thank you for pointing this out. Upon re-examining the structural data, we agree that the side chain orientations and inter-residue distance are not fully consistent with a canonical π - π stacking interaction. We have revised the statement accordingly to clarify that W57 and W162 are in close spatial proximity with a potential for interactions rather than specifying a defined π - π stacking interaction (page 7).

(ii) P8, 1 1. There is no N142; S142 and N143 – in any case, both residues are over 7 Å away from Y60.

Response: Thank you for pointing this out. We apologize for the oversight regarding the residue numbering. Upon re-examination, we confirm that it is N143 (not N142) whose side

chain forms a hydrogen bond with the backbone carbonyl of T58 in Nb20. We have corrected the residue numbering and revised the text accordingly.

(iii) P8, 13, No D198, E198 is far apart from K79 to form a hydrogen bond.

Also, considering these features, it is not clear how mutations in these residues affect the interaction between Nb20 and mGlyR (Supp Fig 6 and 8).

Response: Thank you for pointing this out. Upon re-examination, we confirm that it is E198 (not D198) whose side chain forms a hydrogen bond with N31 of Nb20. We have corrected the residue numbering and revised the text accordingly. Furthermore, we have clarified in the manuscript how mutations at these positions potentially alter the local interaction network (page 8) and thereby influence Nb20 binding to mGlyR, as illustrated in Supplementary Fig 7.

Comment 6. P9, 12, "...In the mGlyR-RGS7/G β 5 structure, the ECD domain could not be observed.... the binding of Nb20 stabilized both the ECD and RGS7-G β 5, enabling the reconstruction of the entire assembly with reasonably well-resolved density."

This is a critical issue as the sentence is not true. The key feature of the authors' model is that Nb20 binds and stabilizes the extracellular domain, which was not visible in Nb20-free state. Subsequently the conformational change is relayed to the TM and intracellular region, which alters the structure of the RGS7-G β 5. In contrast to authors claim, the extracellular domain, TM and RGS7-G β 5 are well defined in the reported Nb20-free mGlyR-RGS7/G β 5 (PDB 7EWP). Because this is the starting point of the authors' model for the structural basis for the Nb20-mediated anti-depressant effect of mGlyR, the authors' proposed model is not consistent with available structures. I strongly suggest that the authors should reinterpret their data by comparing with all available structures including 7EWP.

Response: We thank the reviewer for this important comment. At higher Cryo-EM resolutions, regions of intrinsic flexibility tend to appear more disordered or diffuse in the density map. This is because the electron density reflects the ensemble of conformations sampled by these flexible regions rather than a single static state. While high-resolution maps provide detailed features (such as side chains) for well-ordered, rigid regions, they also make the conformational heterogeneity of flexible domains more evident. Instead of a well-defined density, flexible domains display an averaged-out signal across multiple conformations, resulting in a blurred or "scattered" appearance.

This effect is clearly illustrated in our study through 3D variability analysis, which shows that the extracellular domain (ECD) undergoes significant motion and is highly flexible. Such heterogeneity explains why the ECD appears poorly resolved despite the overall map resolution being high. A similar phenomenon is seen in the previously solved mGlyR structure 7EWP (EMDB 31360) at 4.3 Å resolution: the transmembrane (TM) region shows clear side-chain features, whereas the ECD density lacks side-chain information, indicating intrinsic flexibility in this region even at lower resolution. When structures are refined to higher resolutions—as in our case (3.3 Å)—the TM region becomes even better defined, but

the flexible ECD appears even more scattered due to high-resolution filtering revealing its conformational heterogeneity.

To address the reviewer's suggestion, we have expanded our structural comparison to include PDB 7EWP and other available mGlyR structures. This updated analysis is incorporated in the revised Results (page 8, Supplementary Fig. 8F, H), where we describe the conformational differences observed.

Comment 7. P9. Even in the structure after focused refinement, the EM density for Nb20 are not well-defined. This is observed from the pdb and map from the authors and Supplementary Fig. 7D. Thus, one possibility is that the end-regions of the complex including Nb20 and Gb5 are flexible instead of the Nb20-binding induced release of RGS7-Gb5 from mGlyR. Authors may consider such possibility. Also, the model-fitted-map seems unmatched each other in supplementary Fig. 7D. It would be helpful that authors provide better figure or check their orientations.

Response: We thank the reviewer for this insightful observation. We agree that, even after focused refinement, the EM density for Nb20 is not well defined in our map, consistent with the reviewer's assessment of our PDB, EM map. As suggested, one plausible explanation is that the peripheral components of the complex, including Nb20 and G β 5, exhibit higher flexibility, which contributes to the diffuse density observed. We have now included this possibility in the revised Discussion section (pages 9), noting that the observed density does not necessarily imply complete release of the RGS7-G β 5 module from mGlyR upon Nb20 binding. Regarding the reviewer's comment about Supplementary Fig. 7D (Now Supplementary Fig. 6D), we have carefully re-examined the map-to-model fit and the orientations presented. The figure has been updated with improved orientation and rendering and clearer depiction of the model fitting, ensuring that the map and model are accurately aligned and more easily interpretable. The revised figure is provided as Supplementary Fig. 6D in the updated manuscript.

Comment 8. P 9. "...Even after local refinement and analysis at various contour levels, the RGS domain density remained unresolved...":

Authors may consider to perform structural analysis using a newly generated local map encompassing the 7TM-RGS-G β 5 regions. At the current resolution, the map quality is still poor to support claims that the nanobody induced the unresolved G β 5 region. Please consider increasing the contour level to reduce noise from detergent micelles. If the signal is sufficiently strong, the 7TM-RGS7-G β 5 complex—excluding the unresolved regions—should remain clearly visible. This would help determine whether the unresolved areas are due to localized disorder, rather than overall weak density across the entire RGS7-G β 5 region. Then, the authors should represent these maps accordingly. The current maps shown in Supplementary Fig. 7F appear to be at low contour levels.

Response: We thank the reviewer for these constructive suggestions. We have performed additional local refinements focusing on the 7TM-RGS7-G β 5 regions and generated new locally refined maps to better evaluate the density in these areas. We also examined the maps

at various contour levels, including higher thresholds to minimize noise contributions from detergent micelles. As suggested, we now provide updated figures (revised Supplementary Fig. 6F) where the density corresponding to the well-resolved regions of the 7TM–RGS7–Gβ5 complex remains visible even after increasing the contour level. These analyses confirm that the lack of resolved density for portions of the RGS7–Gβ5 region is likely due to localized conformational flexibility rather than an overall weak signal across the entire module. The revised results sections (page 9) now clarify this interpretation, and the updated maps are represented accordingly.

Comment 9. P9. Considering the weak density of intracellular region and more than half of the EM density for Gb5 is missing, it is difficult to say 5 Å translation of the Gb5 in the presence of Nb20. Description of the conformational change with inaccurate structure could mislead the readers and carefully addressed.

Response: We thank the reviewer for pointing this out. We agree that the intracellular region exhibits weak density and that the EM density for Gβ5 is only partially resolved, which makes it challenging to directly interpret small positional shifts with high confidence. In light of this, we have revised the text to present the 5 Å translation of Gβ5 in the presence of Nb20 more cautiously. Rather than describing it as a definitive conformational change, we now discuss it as a *plausible* movement inferred from the available density, while clearly acknowledging the limitations arising from the incomplete map and flexibility of the intracellular module.

Comment 10. P9. “Nb20 binding disrupts key hydrogen bonds: one between R715 from CTH2 and E172 from RGS7, and another between T602 from the TM domain and D147 from RGS7 (Supplementary Fig. 9H).”

T602 and D147 do not form hydrogen bonds in the 7SHF structure. Authors should check this part. Calculate local resolution and present it as a figure to claim residue-shift. In my opinion, 4 Å resolution is not sufficient to claim the residue shift.

Response: We thank the reviewer for pointing this out. Upon re-examination of the 7SHF structure and our refined maps, we agree that T602 and D147 do not form a direct hydrogen bond. We have corrected this statement in the manuscript. Instead, we now describe the interactions more cautiously without overinterpreting the density at this resolution. At the current ~4 Å resolution, we agree that the evidence is insufficient to support a definitive residue shift; therefore, we have revised the discussion to reflect this limitation (Page 9).

Comment 11. The FSC curve for the locally refined map (Supplementary Fig. 6G) shows an abnormally large bump, suggesting overfitting due to an overly tight mask. One of the binary masks shown in Supplementary Fig. 5 also appears too tight, excluding signal near the edge regions. The authors should consider to repeat the local refinement using a new mask generated from a much larger base mask, for example using ChimeraX.

Response: We thank the reviewer for this helpful comment. We agree that the binary mask used in the initial refinement was too tight, which likely contributed to the bump observed in

the FSC curve of the locally refined map (Supplementary Fig. 5G). Following the reviewer's suggestion, we generated a more generous mask and repeated the local refinement. This adjustment significantly reduced the bump in the FSC curve and yielded more reliable refinement statistics. The revised masks and updated FSC curves are now provided in Supplementary Fig. 5G.

Comment 12. Overall resolution is insufficient to claim the loop shifts or residue shifts (Supplementary Fig. 9D, E, H). The authors are highly encouraged to present model-fitted-maps for all regions separately.

Response: We agree with the reviewer that the local resolution is limited in some regions, and therefore caution is required when interpreting residue- or loop-level shifts. In the revised manuscript, we have toned down our claims regarding side chain or residue movements and instead describe the changes at the level of secondary structure or domain rearrangements. In addition, as suggested, we now provide model-fitted maps for the corresponding regions (Supplementary Fig. 6A, B, C), which allow readers to better assess the local map quality and the basis for our structural interpretation.

Minor comments

1. P7 19, "To improve the map, local refinements using soft mak corresponding to ECD-Nb20"

Typo, refinements > refinements, soft mak > soft mask.

Response: We apologize for the typographical errors and we have corrected them

2. Supplementary Fig. 9C and Supplementary Fig. 9F. Please indicate the position of the stalk domain.

Response: We have now indicated the position of the stalk domain in Supplementary Fig. 8C and Supplementary Fig. 8F for clarity.

3. Are Fig. 3C and 3D in the same view? It would be helpful to match their viewpoints for easier comparison. Additionally, indicating the positions of CDR1 and CDR2 would help readers to interpret the figure.

Response: We have adjusted the views to better illustrate the interactions and have now aligned them as closely as possible to match the perspective shown in Fig. 3C, while still maintaining the key interactions.

Reviewer #1

Comment 1.) Is the Nb20 antibody specific to mGlyR in vivo? The null nanobody controls for off-target effects of the surgery and nanobody domain, but does not rule out promiscuous binding of Nb20 to other targets, which could generate indirect effects on behavior. Since the

authors studied mGlyR knockout mice, are the antidepressant-like behavioral effects of Nb20 affected in mice lacking the expected antigen?

Response: We think that we established specificity of the Nb20 effects beyond reasonable doubt. The behavioral effects that we observe in response to Nb20 administration match very well the phenotypes seen in mGlyR KO mice. In brain slices, Nb20 produces the same effect on excitability seen upon genetic elimination of mGlyR. Finally, we use a very stringent control – the same Nb mutated not to interact with its target mGlyR and see no effects with it. On the sum, this is beyond what is typically done to establish the specificity of a biologic. Unlike small molecules, nanobodies and antibodies are usually very target selective and significant engagement of other targets at low concentration we use amidst extremely high affinity for the intended target are not commonly expected. The experiment proposed by the reviewer to evaluate the effect of Nb20 in mGlyR KO mice is great in spirit but unlikely to be informative. mGlyR KO mice show strong baseline anti-depressant phenotype- the same effect produced by the Nb20 treatment – thus their anti-depressant responses can't be evaluated due to “floor/ceiling” effect. However, we understand that one can never be certain about lack of off-target activities. Thus, we revised the manuscript to indicate this caveat in the Discussion section (page 14).

Comment 2.) In Fig 5, the authors confirm that Nb20-based mGlyR antagonism phenocopies the increased excitability of L2/3 PFC neurons in mGlyR KO mice, which is a valuable control. However, it's not clear how this relates to the behavioral effects observed after Nb20 treatment in Fig 4 since the treatment timescales differ significantly (10 min treatment in vitro in Fig 5 vs 24+ hrs in vivo in Fig 4) and these effects may be correlative. If the authors performed these electrophysiology experiments from mice treated with Nb20 (as in Fig 4), are these neurons more excitable? Alternatively, are proxies of excitability, such as cFos, increased in mice treated with Nb20?

Response: Electrophysiology experiments were designed to test whether inhibition of mGlyR with Nb20 would produce similar effect as mGlyR knockout or its inhibition by glycine establishing the effects at a more mechanistic level. We understand the reviewer's point and certainly agree that recording electrophysiological properties in neurons of mice following Nb20 administration and behavioral evaluation would have been ideal and very powerful. However, these experiments are extremely challenging and difficult to interpret due to several reasons. The main problem and the biggest limitation are the complexity of the stress responses in terms of their effects on the neuronal activity of cell populations and circuits which we found to be highly variable. This is not endemic to our study and in fact the effects of stress on circuits and neuronal activity is a hotly debated area with a range of disparate effects described (e.g. PMIDs: 18923511, 28856337, 34893785). Thus, what may appear as a simple follow up in a context of a Nb20 treatment in fact requires defining stable effects of stress on neural circuits which is a momentous feat even if one looks at “simpler” proxies such as cFos staining. We appreciate reviewer's understanding that such experiments might not be feasible for us to accomplish this in a framework of a study that mostly geared towards atomic/molecular angle of understanding how Nb20 alter mGlyR. We further acknowledged

these limitations and the need to better define physiological effects of Nb20 on neural circuits and neuronal properties in the future studies.

3.) While the results are indeed exciting, the manuscript appears overly hyperbolic in several areas. For example, in the discussion they “report a unique immunotherapy solution for a major neuropsychiatric condition- major depressive disorder”, but depression reflects a human condition with environmental drivers that are not addressed in this manuscript. Additionally, the authors claim immunotherapies offer “low toxicity”, but one of the current limitations of immunotherapies is toxicity, partially due to an incomplete mechanistic understanding (which is highlighted by points 1 and 2 above).

Response: We certainly understand the limitations and arduous task of translating the observations in rodents to humans. We have tried acknowledging these limitations in the manuscript and have done more of this during this revision (e.g. page 13). We also revised the manuscript to subdue hyperbolic claims.

We appreciate reviewer's help in improving our manuscript. We have revised the manuscript to address outstanding issues. Our responses to specific concerns are as follows.

1.. Supple Fig 6 a, c, d, f are still messy and too complicated.

(i) Authors should provide better views or sections of the maps and models. Fig S6f. Even at very low contour level (near background) of the map where micelle completely covers the TMs, very poor density for RGS7 is shown. I do not think it is proper to build the model here. Accordingly, description of the conformation changes in RGS7 is unlikely to be accurate (as density for the secondary structures for RGS7 is unclear) and should be removed.

Response: Based on the suggestions of the reviewer we have improved the view for the TM domain and the RGS7 domains. As demonstrated in our previous revision using 3D variability analysis, both the RGS7 and ECD domains exhibit substantial mobility relative to the transmembrane domain. Consequently, these regions remain at lower local resolution, although their overall positions could still be defined through rigid-body docking and refinement. Moreover, we do observe clear, albeit low-feature, density corresponding to RGS7 and only the global shape profile G β 5 (hence it was rigid body docked into the map). We included them in the model through rigid-body fitting to provide a complete and accurate representation of the complex architecture. The DEP/DEX domain of RGS7, which directly contacts the receptor, is shown in better view, showing the secondary structure quite visible in Supplementary Figure 6. The RGS domain remained invisible in the map and hence was not docked into the maps. We do see garbled density corresponding to G β 5 and hence, it was docked into the map and was not built further due to lack of sufficient information.

(ii) Also, in FigS6f, some TMs do not form helices but rather they form loops (see figure S6f).

Response: This issue arose due to an incorrect interpretation of the PDB file by Chimera, which displayed certain transmembrane helices as loops. We have now corrected this visualization error, and the updated Supplementary Fig. S6f properly shows these regions as continuous α -helices within the well-defined transmembrane domain.

(iii) Method section: Authors described "...manual building..". How RGS7 – G β 5 can be manually built? Authors should more precisely describe what they have done. Also, authors should describe which parts of the models are visible in the maps either in the main text or methods section. In principle, model should not fit in the invisible region.

Response: The Methods section has been revised to more precisely describe the model-building procedure. We now specify that only the well-defined regions of the transmembrane domain were manually adjusted in Coot based on visible cryo-EM density, while G β 5 and RGS7 were docked into the map as rigid bodies. The RGS domain, which is not visible in the map, was not built and remains absent from the final model. We cannot leave the density observed at very low contour levels entirely unaccounted for; therefore, we docked RGS7 and G β 5 into this region based on our prior structural information, which confirms that the corresponding density arises from these components. The revised Methods section explicitly describes that the transmembrane region was built de novo into the density, whereas RGS7 and G β 5 were docked and refined as rigid bodies. Additionally, we have clarified that no part of the model was fitted into regions lacking interpretable density.

2. P7, line 2 and supple Fig 4 & 5,

(i) the resolution is unlikely to be 3.83 Å. In S5D, the GSFSC curve decreases and increases again near 0.143 (unusual type of curve), and this is unlikely to be the true resolution.

Response: We appreciate the reviewer's careful examination of the GSFSC curve. We also noticed the unusual dip in the curve near the 0.143 threshold during multiple steps of data processing. We believe this feature is intrinsic to the dataset and may arise from the non-uniform distribution of particle orientations or conformational flexibility within the complex, the exact reason for this dip is far from understood. Despite this minor anomaly, the overall map quality supports a global resolution of 3.89 Å. The transmembrane domain, in particular, exhibits well-defined density with clearly visible side chains, consistent with this resolution. We note, however, that the RGS and ECD domains are resolved at lower local resolution due to their inherent flexibility, as also indicated in variability analysis.

(ii) Authors wrote 3.83 Å in the text, but 3.89 Å in the figure section – which one is correct? Also 3.43 vs 3.49 ? in S5D.

Response: We thank the reviewer for noticing this inconsistency. The discrepancies in the reported resolutions have been corrected throughout the text and figures for consistency.

(iii) Also, authors should provide the local resolution maps for the structure with different colors. Possibly, they may represent maps including local resolution at the bottom of Fig S4.

Response: The figure has been updated to include local resolution maps rendered in color, illustrating the resolution variation across the structure. The local resolution representation has also been added at the bottom of Supplementary Fig. S4 as requested.

3. The focused refined EM density for the Nb20-ECD is not sufficient to fit the side chains in most of the regions. Therefore, description of the interface interaction could provide wrong information. Some examples include;

(i) E198 – N31; shorter one has 4.5 Å distance, the other one is over 9 Å and both are too far. Moreover, carbonyl - carboxyl oxygen in shorter one

Response: We agree that the focused-refined map for the Nb20–ECD complex does not allow unambiguous placement of all side chains at high confidence because this interface can not be resolved with higher detail in our structures presented. We have therefore moderated our description of the Nb20–ECD interface in the text to avoid overinterpretation and caution. The interaction of E198–N31 have been re-evaluated, and all specific atomic interactions that were not reliably supported by the map have been removed from the manuscript.

(ii) between R56 of Nb20 and E166 of subunit A: density for the R56 side-chain is not visible unless you lower the contour level near the background level.

Response: We acknowledge that the side-chain density for R56 of Nb20 is weak and becomes visible only at lower contour levels. Accordingly, we now note in the text and figure legend that the side-chain positioning in this region is tentative and should be interpreted with caution.

(iii) Nb20* mutant: the mutation study performed here is more likely complete disruption of the local

core structure CDR1 and CDR2 rather than disruption of the interaction by the surface mutation. I do appreciate if authors have attempted to provide clearer mutational effects, but at least brief comment would help.

Response: Regarding the Nb20* mutant, we appreciate the reviewer comment. We agree that the mutations introduced may affect the local core structure of CDR1 and CDR2, potentially influencing Nb20 more globally in addition to the interface interaction. We have added a brief statement discussing this possibility in the revised Results section.

Also, in method section: Nb20*: residue number “29 to 35” vs “30 – 35” in the maintext & others. These numbers are inconsistent number with PDB, which are the correct numbers?

We thank the reviewer for pointing out the inconsistency in residue numbering. The numbering has been carefully cross-checked with the PDB file, and all references are now consistent across the main text, Methods section, and figure legends. The correct residue range for Nb20* is residues 30–35 and has been changed accordingly in the text and figures.

4. P 8 “.. mGlyR– RGS7–Gβ5 (PDB: 7SHF, 7EWP) with that of the Nb20-bound complex. While the overall TM domains are largely conserved, notable deviations of up to 5.5 Å were observed in TM3, TM4, and the connecting loop (Supplementary Fig. 8F, H)...”

I have compared the TMs (TM3, TM4 and other TMs) and linkers between the current model (mGlyR-Nb20-RGS-Focused2.pdb) and 7EWP by aligning entire TMDs, and found out no notable differences. In fact, the largest differences in TM3, TM4 and connecting loops are near or less than 1 Å. Authors made inaccurate description and must correct or delete the sentence.

Response: We thank the reviewer for the analysis and for pointing out the difference observed when aligning the entire TMD. As originally suggested by the reviewer, we have included PDB: 7EWP in our comparison of the transmembrane domains. Upon re-analysis, we confirm that the deviation between the mGlyR–Nb20–RGS–Focused2 model and 7EWP is indeed small (approximately 1.2 Å), consistent with the reviewer’s observation.

However, the reviewer has to also consider the comparison with our previously published structure (PDB: 7SHF). In this comparison, we observe a larger local deviation of up to 5.5 Å, particularly near residue T514 in TM3. This difference reflects subtle but conformational variability among the available mGlyR–RGS7–Gβ5 complexes, rather than any modelling inaccuracy (Supplementary Fig. 8F).” Thus, we disagree with labelling of our interpretation as “inaccurate”.

To ensure accuracy and fairness, we have revised the manuscript to clearly state the range of deviations observed across all available structures (7EWP, 7SHF, and the current model) and to clarify that the higher deviation originates from the comparison with 7SHF.

5. “that this domain becomes highly flexible upon binding the Nb20 to ECD. Superimposition of Nb20-bound and free structures by the 7TM region suggested plausible changes.....that remodeling at the 7TM interface triggers changes in contacting residues of...”

Since there are no noticeable changes in TM, this sentence is not correct. Considering TMD retains same structure upon Nb20 binding, authors should reconsider their proposed model.

Response: We thank the reviewer for the comment. In light of our previously published structure (PDB ID: 7SHF), in addition to comparison with 7EWP, we maintain that the observed flexibility of the domain upon Nb20 binding is supported. However, we agree that the original phrasing could be misleading regarding the transmembrane domain. We have therefore rephrased the sentence to clarify that while the 7TM region remains largely unchanged, binding of Nb20 to the ECD induces local flexibility in the extracellular and RGS domains, which may modulate interactions at the interface.

6. “In the mGlyR-RGS7/Gβ5 structures, the ECD domain was highly flexible, resulting in low resolution features as observed in 7EWP or scattered density at relatively higher resolution in 7SHF structures. The binding of Nb20 appeared to stabilize both the ECD and RGS7-Gβ5 module, yielding relatively well-resolved density for ECD. Strikingly, comparison revealed that the density corresponding to the RGS domain, which typically wraps around Gβ5 from below, was poorly resolved in the Nb20-bound structure, suggesting that RGS domain adopts”

This is not properly revised paragraph. This is only to justify their claims that Nb20 stabilizes ECD. Resolution of the 7EWP ECD is not sufficient to place the side-chain, but main-chain for ECD is well defined. Nb20-bound EM map is no better (to place the side-chain) than the Nb-free map (7EWP) to claim that Nb20 stabilizes ECD of mGlyR.

Response: We agree that the current resolution of both the Nb20-bound and Nb20-free (7EWP) maps does not allow reliable placement of ECD side chains. However, in light of our previously published structures, where the ECD was not observed upon binding the RGS complex, the presence of density for both the ECD and RGS complex under our experimental conditions hints at a potential stabilizing effect of Nb20 on the ECD. This observation differs from the 7EWP structures reported by other groups. Keeping these points in mind, we have revised the paragraph to modify statements suggesting that Nb20 improves ECD stability. The updated text now focuses on the observed map features without over-interpreting the effects of Nb20 binding.

(i) low resolution vs high resolution: Description of resolution is meaningful only if the features are present and can be comparable. This sentence gives impression that the ECD of 7EWP is lower resolution than that of 7SHF. However, 7SHF does not have ECD, a key feature authors described. Comparison of the local resolution of ECDs between 7EWP and 7SHF would be reasonable. Thus, there is no point of describing resolution here, and should be removed as this type of description only confuses the readers.

(ii) For the readers to clearly understand, the references for 7EWP and 7SHF should be included.

Response: We thank the reviewer for the suggestion. References for PDB entries 7EWP and 7SHF have been included in the text to ensure clarity for the readers.

(iii) After careful checking of the cryo-EM map for the ECD in Nb20-free and Nb20-bound, I cannot agree with authors claims that Nb20-binding stabilizes ECD and their further idea.

Response: While we agree that the maps do not allow unambiguous placement of ECD side chains and the original claim was overstated, we do observe density for the ECD in the RGS7-Gβ5-bound complex in the presence of Nb20, which was not observed in our previous structures in presence of RGS7. This contrasts with our previously published structures, where the ECD was not visible when the RGS complex was bound, suggesting a potentially improving the map corresponding to ECD in presence of RGS7-Gbeta5 bound.

Accordingly, we have moderated our claim in the revised text to avoid overstatement. We now state that Nb20 may contribute to stabilizing the ECD in the presence of RGS7–Gβ5, while focusing on the observed map features rather than over-interpreting the effect of Nb20 binding.

7. “..but relatively low resolution in the ECD and Nb20 binding regions. Nevertheless, these maps allowed us to construct the complete models of the mGlyR-Nb20 and mGlyR-RGS complexes,”.

The map for the RGS complex is poorly defined and not sufficient to build the complete model. As I commented above, it is not possible to trace the main-chains for RGS7 and Gb5. Thus, authors should revise this part.

Response: We thank the reviewer for the comment. We agree that the density for the RGS7–Gβ5 region is relatively poorly defined and does not allow manual tracing of main chains. Accordingly, we have revised the text to clarify that the RGS7–Gβ5 module was rigid-body docked and refined into the cryo-EM density rather than manually adjusted in Coot. The observed density corresponds to this region, and we cannot leave it unaccounted for, as its assignment to RGS7–Gβ5 is well supported by our previous structural knowledge. The revised description now accurately reflects the modelling approach used for this flexible region.

8. Table 1 for the statistics for the structural analysis. Why only two PDBs are reported? Are authors not going to deposit the focused-refined structures? Also, authors should clearly describe which data will be available to public with their statistics.

Response: The focused refined structures do not enhance the local resolution significantly for the ECD or the RGS7-GB5 regions which were poorly resolved in the consensus refined structure therefore we have deposited just two PDBs (Nb20-mGlyR, Nb20-mGlyR-RGS7-Gβ5) and the corresponding maps (EMDB-65225, EMDB-65226)

Thank you for evaluating our revised manuscript. We have adjusted the manuscript per guidelines and in response to further reviewer's critique. We hope you find the revised manuscript to be acceptable for publication at Nature Communications. Our response to reviewer's comment is as follows. Modifications in response to reviewer's comment are highlighted in blue.

Comment: "To explain the Nb20's anti-depression effect, authors proposed a model: Binding of Nb20 to the extracellular ligand binding domain of mGlyR promotes specific stabilized conformation of the receptor which propagates through the stalk region to remodel the 7TM region, which then serve as the contact site for RGS7. Authors provided a detailed description in the text; "...remodeling at the 7TM interface triggers changes in contacting residues of the RGS complex. Nb20 binding appeared to shift CTH2 by 2.2 Å, induce a 1.2 Å displacement in RGS7, and 1.5 Å movement in the TM3-TM4 connecting loop, and up to a 5.5 Å (when compared with 7SHF) changes in the TM3, which serve as the contact site for RGS7..."

In previous review, I have commented that comparison of the structures of the 7TMs between the Nb20-bound- and Nb20-free mGlyR-RGS7/Gβ5 (PDB ID: 7EWP) showed less than 1 Å differences in Ca atoms. While authors agreed with my observation on small structural difference in 7TMs, they maintained their claim that the 7TMs has been remodeled by Nb20-binding, and retained their description in P9 in the main-text. I am not convinced on authors' claim. I understand that the authors tried to explain structural differences using their previously published structure alone (PDB ID: 7SHF). But there is other structure available which should be also used for comparison. This is very simple issue. There is Nb20-free mGlyR-RGS7/Gβ5 structure which is determined independently and has been available in database for several years. Comparison with this structure with the current Nb20-bound mGlyR is unbiased. Its 7TMs structure is virtually identical to that of the Nb20-bound mGlyR-RGS7/Gβ5 presented in this work. Can authors claim structural changes in 7TMs, although Nb20-binding does not affect the structure of 7TMs of mGlyR? If so, how would the authors explain the structural identity/similarity of 7TMs between them? Instead of maintaining their claims on conformational changes in 7TMs, authors may consider other possibilities. In my opinion, authors can use Nb20-mGlyR interaction, their strong biological data with animal model, which would make more clear story."

Response: We agree with the reviewer that the remaining issue is simple and essentially deals with the interpretation. In our opinion, structures present ground truth. Extensive review of our work has not identified any issues with the quality of the structure that we present. Thus, the structural information presented in this manuscript can be analyzed by anyone and interpreted in multiple possible ways. In this manuscript, we provide our analysis and interpretations. We have injected significant transparency during the last revision comparing our Nb20 structure with the apo structure we obtained earlier 7SHF and also the one that reviewer favors – 7EWP. The former comparison shows more extensive changes, the latter less. In our opinion, comparisons of the structures determined by the same group using identical conditions for structural work is

more accurate than across groups as conditions, e.g. choice of detergent, etc can influence the structure. With this, the differences in opinion become an interpretational issue and as authors we would like to be able to voice our opinion on the matter.

Nevertheless, we understand reviewer's point and the editorial view and in response we have provided further clarifications and acknowledgement of limitations and possible interpretational differences. Specifically, we adjusted the description of our model on pages 9 and 10 to suppress conclusions about changes in 7TM architecture caused by Nb20 binding. We also acknowledged gaps and limitations very explicitly in the Discussion on page 15 stating the following. *“How exactly Nb20 influences the conformation of 7TM region and particularly its interface with RGS7/Gβ5 has not been fully resolved in this study, given the differences between two previously reported ground state structures of mGlyR without Nb20. Additional structural work that captures intermediate conformations of the receptor complexes will be required to fully resolve the mechanism of RGS/Gβ5 modulation by mGlyR ligands such as Nb20 or glycine.”*

Thank you for your consideration!

Best wishes,

Kirill Martemyanov